



# Seasonal variability of surface and column carbon monoxide over megacity Paris, high altitude Jungfraujoch and Southern Hemispheric Wollongong stations

Yao Té[1], Pascal Jeseck[1], Bruno Franco[2], Emmanuel Mahieu[2], Nicholas Jones[3], Clare Paton-Walsh[3], David W. T. Griffith[3], Rebecca R. Buchholz[4], Juliette Hadji-Lazaro[5], Daniel Hurtmans[6], and Christof Janssen[1]

[1]LERMA-IPSL, Sorbonne Universités, UPMC Univ Paris 06, CNRS, Observatoire de Paris, PSL Research University, F-75005, Paris, France
[2]Institut d'Astrophysique et de Géophysique, Université de Liège, B-4000 Liège, Belgique
[3]Center for Atmospheric Chemistry, Faculty of Science, Medicine & Health, University of Wollongong NSW 2522 Australia
[4]Atmospheric Chemistry Observations & Modelling Laboratory, National Center for Atmospheric Research, Boulder, CO, USA
[5]Sorbonne Universités, UPMC Univ. Paris 06, Univ. Versailles St-Quentin, CNRS/INSU, UMR 8190, LATMOS-IPSL, Paris, France
[6]Spectroscopie de l'Atmosphère, Service de Chimie Quantique et Photophysique, Université Libre de Bruxelles, Brussels, Belgium

*Correspondence to:* Yao Té (yao-veng.te@upmc.fr)

**Abstract.** Carbon monoxide (CO) is an atmospheric key species due to its toxicity and its impact on the atmospheric oxidizing capacity, both factors affecting air quality. The paper studies the altitude dependent seasonal variability of CO at the three different sites Paris, Jungfraujoch and Wollongong, with an emphasis on establishing a link between the CO vertical distribution and the nature

of CO emission sources. The CO seasonal variability obtained from the total columns and from the free tropospheric partial columns shows a maximum around March-April and a minimum around September-October in the Northern Hemisphere (Paris and Jungfraujoch). In the Southern Hemisphere (Wollongong) this seasonal variability is shifted by about 6 months. Satellite observations by IASI-MetOp and MOPITT instruments confirm this seasonality. Ground-based FTIR is demon-

strated to provide useful complementary information due to good sensitivity in the boundary layer. In situ surface measurements of CO volume mixing ratios in Paris and at Jungfraujoch reveal a time-lag of the near surface seasonal variability of about 2 months with respect to the total column variability at the same sites. The chemical transport model GEOS-Chem is employed to interpret our observations. GEOS-Chem sensitivity runs allow identifying the emission sources influencing the seasonal

cycle of CO. In Paris and on top of Jungfraujoch, the surface seasonality is mainly driven by anthropogenic emissions, while the total column seasonality is also controlled by air masses transported





from distant sources. In the case of Wollongong, where the CO seasonality is mainly affected by biomass burning, no time shift is observed between surface and above the boundary layer.

## 1 Introduction

Air is one of the most fundamental prerequisites for life and human beings inhale about 1500 litres of air per day. This air contains, besides of the major gaseous components nitrogen, oxygen and argon, reactive trace gases and small particles that are of concern for human health. The survey and control of these trace components, which affect air quality, have thus become a field of major importance for environmental research and public health authorities, especially in large cities. The quality of air

is a function of time and space, which depends on many parameters such as geographic location, meteorological conditions, as well as sources and sinks of pollutants. It is thus strongly affected by natural and anthropogenic emissions.

Atmospheric carbon monoxide (CO) is an important trace gas, due to its toxicity and its impact on the atmospheric oxidizing capacity and air quality. For example, Sam-Laï et al. (2003) have

studied the seasonal phenomenon of CO poisoning mainly due to defect heating systems. Barnett et al. (2006) have investigated the link between outdoor air pollution and cardiovascular hospital admissions. Levy (2015) has studied the effect of CO pollution on the neurodevelopment. The major sources of CO are fuel and energy related industries, heating, motor vehicle transport, biomass burning, and the secondary oxidation of methane and of volatile organic compounds (VOCs such as

isoprene and terpene), which are emitted by plants. Due to the fast reaction (R1), carbon monoxide is the major sink for the main atmospheric oxidation agent, the hydroxyl radical OH (Weinstock, 1969; Bakwin et al., 1994).

$$CO + OH \rightarrow CO_2 + H \tag{R1}$$

A global increase of atmospheric CO thus leads to a decrease in global OH, which in turn aug-

ments the concentration of other, potentially harmful atmospheric trace gases (Logan et al., 1981; Thompson et al., 1990; Thompson, 1992) or potent greenhouse gases sensitive to oxidation such as methane.

This paper characterizes the CO seasonal variability at three ground-based FTIR sites: Paris megacity, remote Jungfraujoch and Southern Hemispheric Wollongong. These sites have been se-

lected for their representativeness of different environments (remote vs. moderate and high pollution sites, Northern vs. Southern Hemisphere) and meteorological conditions:

- Megacity of Paris (France): A high resolution Fourier transform spectrometer (FTS-Paris) has been installed in 2006 on the campus of "Université Pierre et Marie Curie" in the centre of the French capital Paris (48°50'47"N, 2°21'21"E, 60 m asl). Since then, the instrument



is continuously operated by LERMA.[1] The Île-de-France region covers a surface of about 12000 km$^2$ with more than 11 million inhabitants (16 % of the French population). The region has a relatively flat relief with an average elevation of 108 m above sea level (asl) and is strongly influenced by the Atlantic Ocean. Between 1970 and 1990, levels of ambient CO were quite high and stable around 3.5 ppmv (Joumard, 2003). After 1990, a strong decrease

was observed due to new European regulations for motorized vehicles (91/441/EEC Council Directive) becoming effective in the same year. Since 2008, the CO level in the Paris region has been quite stable with no significant trends.

      – Jungfraujoch (Switzerland): The International Scientific Station of the Jungfraujoch (ISSJ) is located in the Swiss Alps (46°33'N, 7°58'48"E, 3580 m asl). Two FTIR instruments have been

used at that site, a homemade FTIR from 1984 to 2008 and a commercial Bruker IFS 120 HR from the early 1990 to present, providing one of the longest observational time series for a variety of atmospheric gases. University of Liége (Belgium) is responsible for the operation of the infrared instruments at Jungfraujoch. Due to its elevation, the site primarily probes the European free troposphere and layers above.

– Wollongong (Australia): The station is located in the Southern Hemisphere at Wollongong University (34°24'22"S, 150°52'44"E, 30 m asl). The instrument is operated by the University of Wollongong and provides data since 1996.

All three ground-based FTIR (Fourier Transform Infrared) spectrometers are part of NDACC (Network for the Detection of Atmospheric Composition Change) and/or TCCON (Total Carbon Col-

umn Observing Network) networks and monitor the concentration of CO for several years. Here, we present NDACC data on the seasonal variations of CO and compare the results from the different sites. These remote sensing measurements are compared with results from the satellite instruments IASI-MetOp (Infrared Atmospheric Sounding Interferometer (Tournier et al., 2002)) and MOPITT (Measurements Of Pollution In The Troposphere (Drummond and Mand, 1996)). With respect to

satellite measurements, ground-based FTIR instruments are more sensitive to the boundary layer and can therefore provide complementary data which we compare with surface in situ measurements. Using custom GEOS-Chem model (Goddard Earth Observing System - chemical transport model (CTM, Bey et al. (2001)) simulations, we investigate the impact of local sources on the lower partial column and its variability as compared to the total column.

The paper is structured as follows. In section 2, the different ground-based and satellite instruments will be described. Section 3 presents the measurement data, which are then discussed and compared in section 4.

---

[1]Laboratoire d'études du Rayonnement et de la Matière en Astrophysique et Atmosphères



## 2  Instrument description

### 2.1  Instrumentation at Paris, France

The Fourier transform spectrometer (FTS-Paris) is a model IFS 125 HR Michelson interferometer from Bruker Optics, cf. http://www.bruker.com. Its maximum optical path difference is up to 258 cm, which corresponds to a spectral resolution of $2.4 \times 10^{-3}$ cm$^{-1}$. The instrument is equipped with IR optical elements ($CaF_2$ entrance window and beamsplitter, InSb detector), suited for ground-based atmospheric observations (Té et al., 2010). Solar absorption measurements are achieved by

coupling the FTS-Paris instrument to a sun-tracker (model A547 from Bruker Optics) installed on the roof terrace. The solar disk is tracked with an accuracy of less than 1 arcmin. The spectra contain rovibrational signatures of many atmospheric constituents, including numerous atmospheric pollutants. The spectral range determined by the above choice of optical elements and detectors is limited to the range between 1 and 5.4 $\mu$m. It is further narrowed down using appropriate band pass filters in

order to optimise the signal to noise ratio when focussing on specific target gases. For CO, the chosen optical filter and the InSb detector allow to cover the spectral domain from 3.8 to 5.1 $\mu$m, which corresponds to a typical NDACC configuration. More instrumental details and different measurement configurations are specified elsewhere (Té et al., 2010, 2012).

Continuous in situ measurements of the CO surface concentration are performed using a commer-

cial CO11M analyser (Environnement SA). The operating principle of the CO analyser is based on the CO infrared absorption at 4.67 $\mu$m, which is the same spectral band covered by the FTS-Paris. Ambient atmospheric air is drawn from the building rooftop into the analyser via PTFE tubing using a diaphragm pump, which is limited to a gas flow of 80 litres per hour. The pumped air is analysed in a 20 cm length multi-path absorption cell with an absorption path length of 5.6 m, using a globar

IR source and a photoconductive PbSe detector. The CO11M analyser has a sensitive range between 0.1 and 200 ppmv, with an uncertainty of 50 ppbv for each individual measurement. Recorded values are time averages over 15 minutes.

### 2.2  Instrumentation at Jungfraujoch, Switzerland

The Jungfraujoch station in Switzerland is currently equipped with a Bruker IFS 120 HR, which is

part of the NDACC network. A thorough description of the instrumentation is given in Zander et al. (2008). Infrared solar spectra are recorded under clear-sky conditions and, thanks to the high altitude, the interference by water vapour is significantly limited in these observations. The instrumental setup is similar to the one used in Paris, although spanning here the 4.4 to 6 $\mu$m range. The spectra are recorded with an optical path difference alternating between 114 and 175 cm. The integration time is

either 135, 404 or 1035 s, corresponding to 3 or 9 scans of 45 s, or 15 scans of 69 s. High resolution observations are only recorded under slowly varying geometry, i.e. for zenith angles lower than $\sim 70^{\circ}$.





### 2.3 Instrumentation at Wollongong, Australia

The Wollongong instrument is also an IFS 125 HR Michelson interferometer from Bruker Optics.
Using the NDACC mode, CO total and partial column data are produced using 3 micro-windows
from the $4.6\,\mu$m CO band (Zeng et al., 2015). The instrument setup is also similar to the Paris and
Jungfraujoch spectrometers, with an optical path difference of 257 cm, with two spectra co-added
for an integration time of 206 seconds.

Measurements of surface CO at Wollongong are combined from two high-precision in situ FTIR
trace gas analysers (Griffith et al., 2012). The analysers use an IR source, modulated through a
Michelson interferometer with a $CaF_2$ beamsplitter. The modulated IR beam is passed through a
dried atmospheric sample within a White cell in a 24 metre folded-path and subsequently detected
by thermoelectrically-cooled MCT (Mercury Cadmium Telluride) detector. Ambient air is measured
daily over 23.5 hours, with 30 minutes reserved for calibration using constant composition air. Am-
bient air is flushed through an inlet line at 5 L/min and sample air is continuously drawn from this
line through the instrument at 1 L/min. The SpectronusTM software (Ecotech P/L, Knoxfield, VIC,
Australia) is used to automate internal valve control and stabilise parameters, such as flow, pressure
and temperature. Recorded spectra are averaged over 3 minutes. Non-linear least-squares fitting of
CO occurs in two broad spectral regions (from 4.33 to $4.65\,\mu$m and from 4.46 to $4.76\,\mu$m), using
the program MALT (Multiple Atmospheric Layer Transmission, Griffith (1996)). Data are reported
as dry-air mole fraction, with a total relative measurement uncertainty below 1%. Wollongong CO
measurements were first analysed by Buchholz et al. (2016) and are publicly available as 10 minute
averages via PANGAEA (doi:10.1594/PANGAEA.848263).

### 2.4 Satellite instruments

The IASI Michelson interferometer (Infrared Atmospheric Sounding Interferometer, (Tournier et al.,
2002; Blumstein et al., 2004) is flying on-board the Meteorological operation (MetOp) polar orbit
platform. The first platform (MetOp-A) was launched on October 19, 2006 and data have been
provided operationally since October 2007. It operates at an altitude of around 817 km on a sun-
synchronous orbit with a 98.7º inclination to the equator. It overpasses each region twice a day.
The MetOp platform has a swath of 30 views of 50 km by 50 km comprising four off-axis pixels of
12 km diameter footprint each at nadir. A second platform (MetOp-B) was launched in September
2012 and the launch of the third and last platform (MetOp-C) is scheduled in October 2018. IASI
observations provide an important contribution to the monitoring of atmospheric composition over
time (Clerbaux et al., 2009).

The MOPITT instrument (Drummond and Mand, 1996; Deeter et al., 2004) is on-board the
NASA's Terra spacecraft in a sun-synchronous polar orbit at an altitude of 705 km. The satellite
was launched on December 18, 1999. MOPITT has been operational since March 2000. The instru-





ment uses the technique of gas-filter correlation radiometry based on the IR absorption bands of CO
to retrieve the vertical profiles of CO. The horizontal footprint of each MOPITT retrieval is 22 km
by 22 km.

In order to be compared to the ground-based FTIR data, satellite data were selected when they are
located in a 30 km × 30 km square centered at the site location: 0.15º around the site latitude and
0.23º around the site longitude.

## 3  Data Analysis

### 3.1  Column data from Paris

Solar spectra were recorded within 3 min at the maximum spectral resolution of $0.0024\,cm^{-1}$. Only
clear sky spectra were admitted to analysis. Available solar spectra cover the time period from May
2009 to the end of 2013, with only very few spectra (about 400 spectra during 19 measurement days)
for the period between 2009 and 2010. Between 2011 and 2013, spectra acquisition became more
regular and more than 4500 spectra from 117 measurement days were recorded and analysed. The
absorption lines of each atmospheric species observed in the solar spectra are used to retrieve its
abundance in the atmosphere by appropriate radiative transfer and inversion algorithms (Pougatchev
and Rinsland, 1995; Zhao et al., 1997; Hase et al., 2006). We have used the PROFFIT algorithm
developed by F. Hase to analyse the Paris data using HITRAN 2008 (Rothman et al., 2009) as
spectral database. PROFFIT is a code especially adapted for the analysis of solar absorption spectra
from the ground and it has been widely applied and tested (Hase et al., 2004; Duchatelet et al.,
2010; Schneider et al., 2010; Té et al., 2010; Viatte et al., 2011). For the retrieval of CO, we have
selected two micro-windows. The $2110.4 - 2110.5\,cm^{-1}$ micro-window is centred around the weak
$^{13}CO$ R(3) line, which is more sensitive to CO at higher altitudes and the $2111.1 - 2112.1\,cm^{-1}$
micro-window around the strongly saturated $^{12}CO$ P(8) line. The left and right wings of that line are
particularly sensitive to CO in the Planetary Boundary Layer (PBL). The retrieval uses a grid with 49
altitude levels. This corresponds to a much thinner atmospheric layering than the effective vertical
resolution indicated by the averaging kernels (Rodgers, 1990). Figure 1. shows that the retrieval
of CO essentially provides two independent measurement points of CO in the troposphere: the
first point delivers maximal information in the altitude range between 0 and 1000 m and thus well
represents the PBL. The second one is representative of the upper troposphere, with a maximum
around 8-9 km. The uncertainties in the CO column density and the profile stem from a variety of
sources. These sources have been investigated in detail by Té et al. (2012), following the procedure
outlined by Rinsland et al. (2000). According to this evaluation, the random uncertainty is around
2.5%.





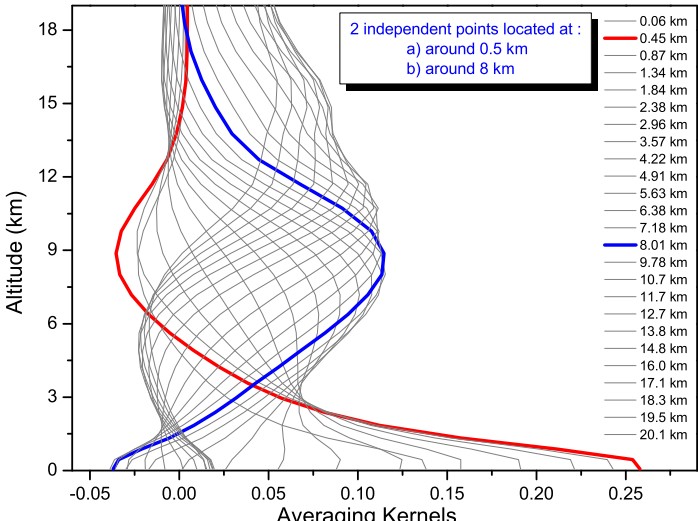

**Figure 1.** CO averaging kernels for each altitude of the a priori profile (from 0.06 to 20 km) using both micro-windows (Paris site).

### 3.2 Column data from Jungfraujoch

The Jungfraujoch data set corresponds to an update of the CO time series described in (Dils et al., 2011). It covers here the January 2009 to December 2013 time period and includes 1733 individual spectra recorded on 539 different days. Mean signal-to-noise ratio (S/N) is 2930, with the 2nd percentile still above 1000. We used the SFIT-2 (v3.91) algorithm (Rinsland et al., 1998) which is based on the semi-empirical implementation of the Optimal Estimation Method (OEM) of Rodgers (1990), allowing retrieval of information on the vertical profile of most FTIR target gases. The standard NDACC approach for the CO retrieval is adopted, fitting simultaneously three micro-windows spanning the $2057.7 - 2058.0$, $2069.56 - 2069.76$ and $2157.3 - 2159.15$ cm$^{-1}$ intervals. The line parameters correspond to the standard release of HITRAN 2004 (Rothman et al., 2005), including the August 2006 updates (e.g. Esposito et al., 2007). The a priori mixing ratio profiles for all interfering molecules (main telluric absorptions by $N_2O$, $O_3$, $H_2O$ and $CO_2$) correspond to a mean of the 1975-2020 version 4 simulation performed for the Jungfraujoch by the WACCM model (the Whole Atmosphere Community Climate Model; https://www2.cesm.ucar.edu/working-groups/wawg). The CO a priori vertical distribution combines WACCM results above 15.5 km, ACE-FTS occultation measurements between 6.5 and 15.5 km (version 2.2, Clerbaux et al., 2008) and extrapolation of ACE-FTS



data down the station altitude, ending at 137 ppbv in the first retrieval layer ($3.58 - 4.23$ km). Additional retrieval settings include a S/N ratio of 150 for inversion, the a priori covariance matrix, with diagonal elements close to 30%/km in the troposphere and extra-diagonal elements computed assuming a Gaussian inter-layer correlation half-width length of 4 km. Objective evaluation of the resulting typical information content indicates that 2 independent pieces of information are available (DOFS of 2.2 on average). The second eigenvector provides some vertical resolution and the determination of partial columns below and above 7.18 km is only marginally impacted by the a priori (corresponding eigenvalue of 0.92, see Fig. 2 of Barret et al., 2003, showing the shape of the three leading eigenvectors). Typical random uncertainties have been evaluated at 2-3% for the total columns and 5% for the $3.58 - 7.18$ km partial columns.

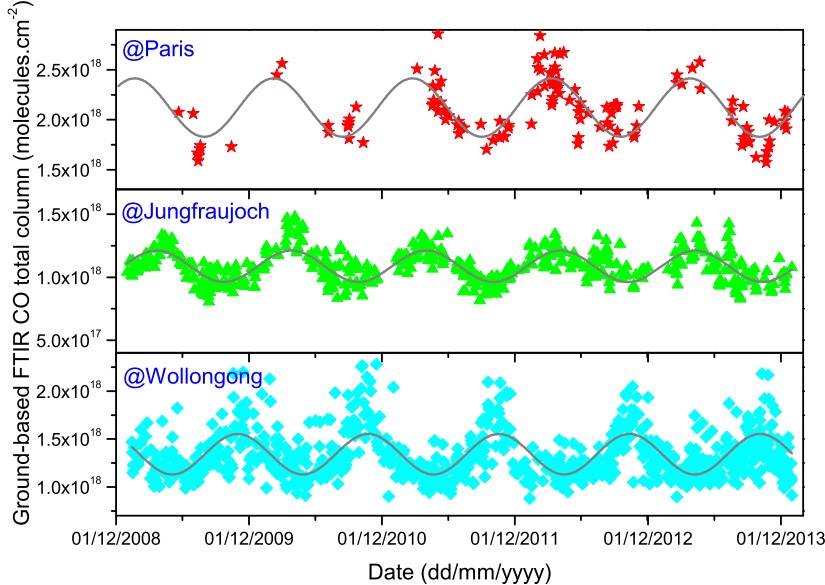

**Figure 2.** CO total columns retrieved by ground-based FTIR instruments at Paris (top), Jungfraujoch (middle) and Wollongong (bottom). Gray lines present CO seasonal variability at each station fitted with sine functions.

### 3.3 Column data from Wollongong

The analysis of the Wollongong NDACC data follows very closely the method described above in section 3.2 for the Jungfraujoch. The algorithm used was SFIT4 v9.4.4 (https://wiki.ucar.edu/display/sfit4/Infrared+Working+Group+Retrieval+Code,+SFIT), an updated version of SFIT2 used in the Jungfraujoch analysis. SFIT4 has inherited the same forward model and inverse method but



with a number of enhancements (not required in the CO analysis), and for the CO retrieval gives the same result. For the Wollongong data, HITRAN 2008 was adopted (Rothman et al, 2009), the mean of the 1980-2020 WACCM version 4 run used as the a priori CO profile (and a 4 km Gaussian interlayer correlation), with the a priori covariance matrix set to 1 standard deviation of the WACCM profiles. A measurement signal to noise ratio of 200 was assumed. This gave a mean DOFS of 2.7. The version 4 WACCM profiles were also used for the a priori profiles of all actively fitted inferring gases ($O_3$, $H_2O$, $N_2O$, $CO_2$, etc.). The error analysis used a NDACC community Python tool to estimate errors assuming a solar zenith angle of 50.2°, representing the mean zenith angle for all Wollongong spectra. The resulting CO total column random errors were calculated to be 2.2%.

### 3.4 Data from the satellite instruments

The IASI-MetOp is a Fourier transform spectrometer with a medium spectral resolution of 0.5 cm$^{-1}$ and a radiometric noise of about 0.2 K at 280 K using nadir viewing and working in the thermal infrared (TIR) range extending from 645 to 2760 cm$^{-1}$ with no gaps. The CO products (L2) from the IASI sounder on the MetOp satellite are downloaded from the ETHER database, cf. http://www.pole-ether.fr, for the period from 1 January 2009 to 31 December 2013. The total column data were generated from the IASI radiance spectra in the 4.7 $\mu$m spectral range and from IASI L2 meteorological data (surface and vertical profile of temperature, humidity vertical profile and cloud cover) (August et al., 2012), using the Fast Optimal Retrievals on Layers for IASI (FORLI) code (Hurtmans et al., 2006). The CO total columns were compared to other CO satellite data (George et al., 2009), from which a relative uncertainty between 4% and 10% could be estimated. The total columns are calculated from the ground altitude to 60 km height. For this paper, we have also additional vertical volume mixing ratio (VMR) profile and partial columns in the PBL and in the troposphere layers around Île-de-France; as well as the partial columns above 4 km height around the Jungfraujoch site.

The MOPITT data were downloaded from the NASA website, cf. https://eosweb.larc.nasa.gov/datapool. We are using the available data of the version 6 retrievals of CO vertical profiles and total columns, for the period from the beginning of 2009 to the end of 2013. The MOPITT retrieval history can be found at the link https://www2.acd.ucar.edu/mopitt/products. Since version 5 of the MOPITT retrieval algorithm, TIR (4.7 $\mu$m) radiances are combined with the near IR (2.3 $\mu$m) daily radiances to improve the sensitivity to lower tropospheric CO over land. The retrieved vertical VMR profile is reported on 10 pressure levels (at the surface and every hundred hPa between 900 and 100 hPa). The retrieved CO total columns are obtained by integrating the retrieved VMR profile. In this paper, we are using the level 2 TIR/NIR products.



### 3.5 Data from the GEOS-Chem model

The global 3-D chemical transport model GEOS-Chem (version 9-02: http://acmg.seas.harvard.edu/geos/doc/archive/man.v9-02) allows for simulating global trace gas (more than 100 tracers) and aerosol distributions. The model is driven by the Goddard Earth Observing System v5 (GEOS-5) assimilated meteorological fields from the NASA Global Modeling Assimilation Office (GMAO), which are at a native horizontal resolution of $0.5^{\rm o} \times 0.667^{\rm o}$. The GEOS-5 data describe the atmosphere from the surface up to 0.01 hPa with 72 hybrid pressure-$\sigma$ levels, at a 6 h temporal frequency (3 h for surface properties and mixing depths). In this study, we use the degraded GEOS-5 meteorological fields as model input to a $2^{\rm o} \times 2.5^{\rm o}$ horizontal resolution and 47 vertical levels, lumping together levels above $\sim 80$ hPa. We apply here the standard full chemistry GEOS-Chem simulation, including detailed $O_3 - NO_x$ – Volatile Organic Compound (VOC) – aerosol coupled chemistry (Bey et al. (2001); Park et al. (2004); with updates by Mao et al. (2010)).

Tropospheric CO is emitted from anthropogenic, biomass burning and biofuel burning sources, as well as from the degradation of many VOCs. The emission inventory of the emissions database for Global Atmospheric Research (EDGAR; http://edgar.jrc.ec.europa.eu) v3.2 (Olivier and Berdowski, 2001) is the global reference for anthropogenic emissions of CO, $NO_x$, $SO_x$, and $NH_3$. For global anthropogenic sources of Non-Methane VOCs (NMVOCs), GEOS-Chem uses the REanalysis of the TROpospheric chemical composition (RETRO; http://gcmd.gsfc.nasa.gov/records/GCMD_GEIA_RETRO) emission inventory (Schultz et al., 2007) for the base year 2000. However, these global inventories may be overwritten by regional emission inventories such as over Europe, where the anthropogenic emissions of CO, $NO_x$, $SO_x$, $NH_3$, propene, acetaldehyde, methyl ethyl ketone and higher C3 alkanes are provided by the European Monitoring and Evaluation Programme (EMEP; http://www.ceip.at) regional inventory for the year 2010 (Benedictow et al., 2010). All these global and regional inventories are scaled to the years of interest according to the method described by van Donkelaar et al. (2008). Anthropogenic sources of ethane and propane are derived from an offline simulation (Xiao et al., 2008). The global biomass burning emissions are provided by the Global Fire Emissions Database (GFED) v3 (van der Werf et al., 2010) and the global biogenic emissions are obtained with the Model of Emissions of Gases and Aerosols from Nature (MEGAN) v2.1 (Guenther et al., 2006)). Methane concentrations in GEOS-Chem are based on measurements from the NOAA Global Monitoring Division flask measurements.

The GEOS-Chem data set employed in the present work covers the period from January 2009 to May 2013 and is derived from a July 2005 to May 2013 simulation, for which the GEOS-5 meteorological fields are available. A 1-year run preceding this simulation was used for chemical initialization of the model. The model outputs consist of CO VMR profiles simulated at the closest pixel to each station and saved at a 3 h time step. The vertical resolution and the sensitivity of the FTIR retrievals have been taken into account for the comparisons involving GEOS-Chem results: the individual VMR profiles produced by the model have been first regridded onto the vertical layer-





scheme adopted at each station, then daily averaged and finally smoothed by convolution with the FTIR averaging kernels (AVKs) according to the formalism of Rodgers and Connor (2003). The regridding method used here is a mass conservative interpolation that preserves the CO total mass

simulated above the altitude of the station (the CO mass below is ignored). The AVKs employed for smoothing are seasonal averages (over March – May, June – August, September – November and December – February, respectively) derived from the individual retrievals of the 2009 – 2013 FTIR data sets.

### 3.6 Data from in situ surface analysers

At Paris, daily in situ surface CO measurements are available for the whole period between beginning of 2009 and end of 2013. At Wollongong, we calculate monthly averages of the in situ data, covering the period from June 2012 to May 2013. Swiss in situ surface data are from the Swiss National Air Pollution Monitoring Network (NABEL), which is a network of 16 observation sites distributed throughout Switzerland in order to measure and record long-term measurement series of

air pollutants. The NABEL monitoring network is operated by EMPA (Air Pollution / Environmental Technology Department). The monitoring stations are representative of different pollution levels. The monthly averaged data were obtained from the annual reports published by the Swiss OFEV (Office fédéral de l'Environnement, http://www.bafu.admin.ch/publikationen/00016). For the paper, we have focussed on the urban sites (Bern, Lausanne, Lugano and Zürich) and the remote mountain

site of Jungfraujoch.

## 4 Seasonal variability

### 4.1 Remote sensing observations

Figure 2 shows the CO total columns of the three ground-based FTIR sites from 2009 to the end of 2013. The data from Paris are less numerous as compared to the other two sites, because measure-

ments are not yet fully automated and acquisitions are only launched when clear sky is expected for more than half of the daytime. Moreover, from 2009 to 2010, Paris CO spectra were recorded only during intensive measurement campaigns, and not on a regularly basis. As expected, the CO abundance is higher in the Northern Hemisphere. The CO mean value is about $2.1 \times 10^{18}$ molecules/cm$^2$ at Paris which is almost twice as high as the value at Wollongong ($1.3 \times 10^{18}$ molecules/cm$^2$). The

CO mean value of $1.1 \times 10^{18}$ molecules/cm$^2$ at Jungfraujoch is quite low due to the site's elevation; the low altitude layers with the highest concentrations of CO that contribute at the other sites cannot do so here.





All three panels show clearly the seasonal variability of CO. We have used a sine function Eq. (1) to characterize this seasonal variability.

$$y = y_0 + A \sin\left(\pi \frac{t - t_c}{w}\right) \qquad (1)$$

where y represents the abundance of CO (in total or partial columns or volume mixing ratio); $y_0$ is the mean value (offset); $A$ and $w$ are respectively the amplitude and the half-period of the seasonal cycle (assumed to be sinusoidal); $t$ and $t_c$ the date and the phase shift in days. Table 1 summarizes the fitted $w$ and $A$ obtained at the three sites.

For the Northern Hemisphere (Paris and Jungfraujoch), the maximum is observed around March-April and the minimum around September-October. The amplitude of the seasonal variability is about $(14 \pm 2)\%$. The average half-cycle is about $184 \pm 4$ days. For Paris, the value ($w = 191$ days) is slightly, but not significantly higher, probably due to the lack of data before 2011. This seasonal variability is also observed by Rinsland et al. (2007) at Kitt Peak, which is the US National Solar Observatory at 2.09 km altitude located in the Northern Hemisphere, by Barret et al. (2003) at the Jungfraujoch, and by Zhao et al. (2002) for Northern Japan. Our observations also agree with a recent 11 years climatology on purely tropospheric CO columns at Northern hemispheric sites (Zbinden et al., 2013), where observed maxima fall within the period from February to April. In the Southern Hemisphere, we observe an expected shift of 6 months as compared to the Northern Hemisphere, with a maximum in October and a minimum in April. We also note that the relative amplitude of the seasonal variation is slightly higher at Wollongong (16% as compared to 14% at Paris), but still within error bars. Interestingly, the relative amplitude is lowest at Jungfraujoch, where the impact of the local surface emissions is small.

The seasonal variability of CO is also observed by the satellite IASI-MetOp and MOPITT instruments, cf. Fig. 3 and Table 1. One of the advantages of the satellite measurements is their spatial coverage. In general, the period and the amplitude of the seasonal variability obtained from the satellite data agree with the corresponding values from the ground-based FTIR measurements. For the high altitude site Jungfraujoch, the satellite data need to be recalculated in order to correspond to the column between the site altitude and the top of the atmosphere (Barret et al., 2003), because the large satellite footprint does not only include the site, but also neighboring areas of lower altitude. Concerning the IASI-MetOp data, the contributions from levels below the Jungfraujoch altitude have been subtracted from the total columns. For the MOPITT data, we extracted the retrieved CO profile for Jungfraujoch and interpolated between the ten pressure levels for a better vertical resolution. MOPITT measurements are performed for specific ground altitudes which, however, are not made available. We have assumed a ground altitude of about 1100 m which is the mean altitude for the Bern canton,[2] to which the Jungfraujoch site belongs. Partial columns above 1100 m were then calculated using the interpolated CO vertical profiles and daily NCEP meteorological pressure

---

[2]https://lta.cr.usgs.gov/GTOPO30 (Global 30 Arc-Second Elevation)



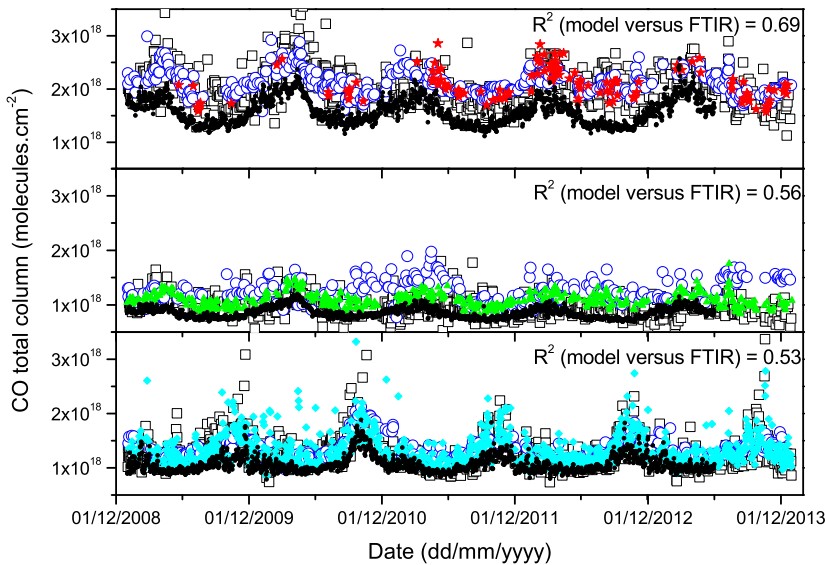

**Figure 3.** Time series of CO columns from satellite instruments and ground based FTIR. Total columns from IASI-MetOp (black open squares) and MOPITT (blue open circles) for Paris (top panel) and Wollongong (bottom panel); partial columns for Jungfraujoch (middle panel). Ground-based FTIR CO total columns (red stars, green triangles and cyan diamonds for Paris, Jungfraujoch and Wollongong, respectively). GEOS-Chem CO total columns are shown as black small full circles.

and temperature profiles. Both interpolated satellite partial columns of CO are plotted in Fig. 3 for Jungfraujoch. Data from the ground-based FTIR instruments and the satellites are in good agree-

ment. This is demonstrated in Fig. 4, where the satellite data are plotted against the ground-based measurements. The good agreement is indicated by robust fits yielding slopes of 0.98 for Paris, 0.91 for Jungfraujoch and 0.99 for Wollongong. The robust fit regression is based on a process called iteratively reweighted least squares (Street et al., 1988). The robust fitting method is less sensitive than ordinary least squares to large changes in small parts of the data.

GEOS-Chem model outputs are presented in Fig. 3 over the entire period from the beginning of 2009 until June 2013. The model is in good agreement with ground-based observations (reasonable correlation with $R^2$ values of up to 0.69), even if the observed total atmospheric CO abundance is underestimated at all three sites: the relative deviations are commensurate: $-24\%$ for Paris, $-21\%$ for Jungfraujoch, and $-20\%$ for Wollongong. These deviations are consistent with previous inverse

modeling studies (Kopacz et al., 2010; Hooghiemstra et al., 2012) and could originate from an underestimation of the emissions of CO and of its VOC precursors in the inventories currently imple-





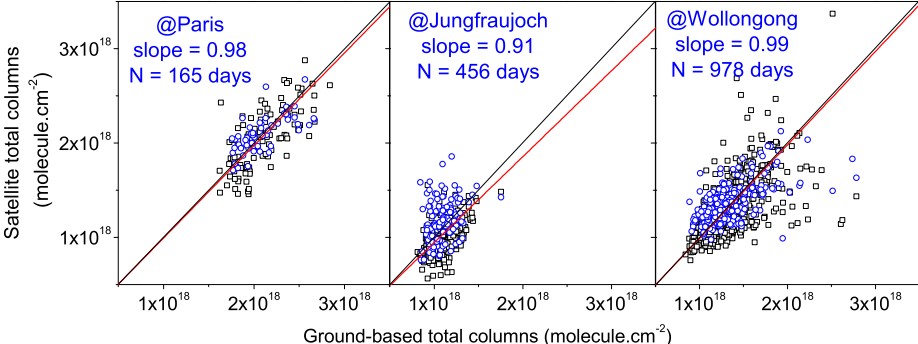

**Figure 4.** Correlation between satellites (IASI and MOPITT) and ground-based FTIR total columns at the three different sites, IASI data are in black squares and MOPITT data in blue circles. Slope values are obtained using robust linear regression using all data. Fitting satellites individually does provide similar results.

mented in GEOS-Chem. Nonetheless, exploring this discrepancy was beyond the scope of this paper that aims at studying the seasonal variability of CO and not at reproducing observed concentrations. It has thus to be underlined that the model shows the same seasonal variability as the measurements:
GEOS-Chem simulations reproduce the Northern Hemispheric maximum in March-April and the minimum in September-October. Therefore, the model is appropriate for diagnosing the seasonal variability. Both, the period and the relative amplitude of the variability are comparable to the measurement results, cf. Table 1. The lower correlation factor between GEOS-Chem and ground-based FTIR for Jungfraujoch and Wollongong as compared to Paris, are probably due to the more complex
orography at these two sites: Jungfraujoch is located in the highest Swiss Alps and the surroundings show very large differences in altitude; Wollongong is sandwiched between the ocean (Tasman sea) and a hilly region (Blue mountains) with a typical altitude of a few hundred meters

### 4.2 In situ measurements of surface CO

Daily averages of the surface CO concentration during the $2009 - 2013$ period at Paris are plotted in
Fig. 5 (bottom panel) in dark green crosses. Days with strong local influence, indicated by a VMR greater than 1 ppmv, are excluded from the analysis. The figure shows a clear seasonal variability with a maximum around January-February and a minimum around July-August. The amplitude of the seasonal variation is about 30%, which is higher than the total column variability. This shows the stronger and direct influence of the local CO emission due to the anthropogenic activities, which
are expected to be particularly high in a megacity. As mentioned in Sect. 3.1, the averaging kernels indicate a good sensitivity of the FTS-Paris instrument to the PBL. The magenta stars in Fig. 5 (bottom panel) represent the averaged CO VMR obtained by the remote sensing measurements for the altitude range between the ground (60 m height) and the 1000 m level. The remote sensing mea-





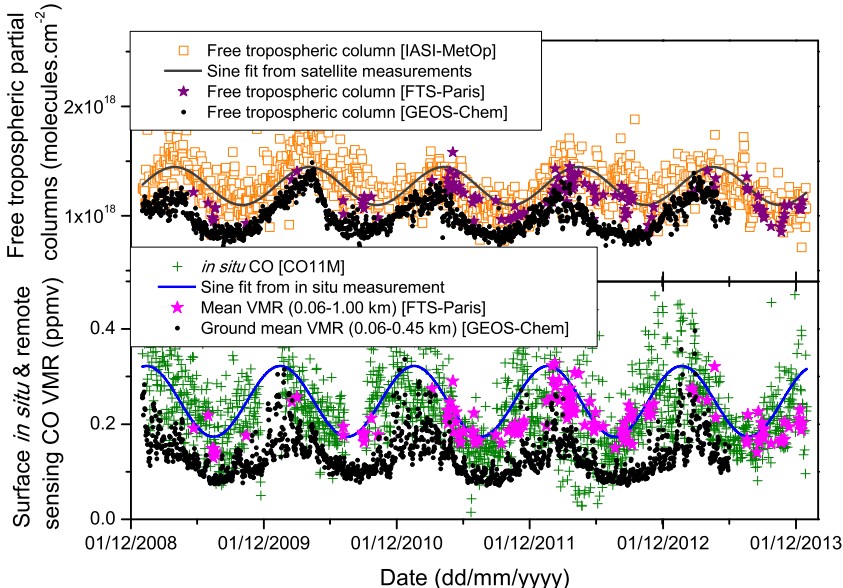

**Figure 5.** Correlation between GEOS-Chem model and ground-based FTIR total columns at the three different sites.

surements are consistent with the in situ data, even if they are much less affected by local pollution
peaks. By comparing Figs. 2 and  5, we notice that the seasonal variability of the total column is
shifted by about 2 months as compared to the variability at the surface. In order to study the free tro-
pospheric columns, we have recalculated the partial columns of CO between 2 and about 12 km over
Paris, obtained by the ground-based FTS-Paris and the satellite IASI-MetOp instruments. Figure 5
(top panel) compares these free tropospheric partial columns with the output from GEOS-Chem.
The seasonal variability in the free troposphere obtained by the three different kinds of data is also
shifted by two months as compared to the surface seasonal variation. As the average lifetime of CO
is estimated to be about two to three months, the seasonal variation of the CO in the atmosphere
is not only due to local emissions, but also due to other natural and anthropogenic contributions:
biomass burning, long distance transport, chemical processes, i.e. oxidization of methane. The sur-
face seasonal variability is directly influenced by the local emission due to human activities: fossil
fuel combustion, warming system, and industrial activities. In comparison the total column seasonal
variability is additionally influenced by distant sources transported to the upper levels of the atmo-
sphere. At Paris, the seasonality introduced by these distant sources outweighs the contribution of
the local surface. The surface CO maximum in January-February corresponds to the winter season,
where domestic heating is strong, where the PBL height is reduced and when oxidation by OH





is lowest. The minimum in July-August corresponds to the summer vacation season during which Paris inhabitants usually leave the city (by more than 50%, http://www.insee.fr), leading to a drastic decrease of vehicle traffic. In order to check the consistency of the GEOS-Chem model, we have plotted the GEOS-Chem surface VMR in the bottom panel of the Fig. 5. The model confirms the

time shift between surface and total column seasonal variability, with a maximum at end of January-February and a minimum at end of July-August. We once again notice an underestimation of the surface CO VMR by the GEOS-Chem model. The discrepancy of about $-37\%$ is larger than the difference of $-24\%$ between GEOS-Chem and ground-based total columns, and can probably be attributed to strong local emissions, which are not in the current emission inventories implemented

in GEOS-Chem.

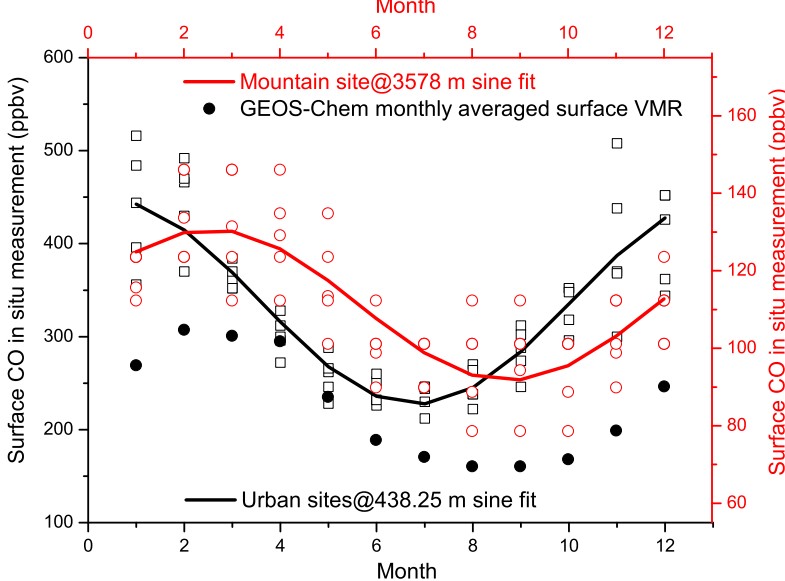

**Figure 6.** CO VMR in the PBL (bottom panel) from surface in situ (dark green crosses) and FTS-Paris (magenta stars) data, and from GEOS-Chem model (black circles). Free tropospheric CO columns (top panel) were calculated between 2 and 12 km, from IASI-MetOp (orange squares) and FTS-Paris (purple stars) measurements as well as from GEOS-Chem (black circles) at Paris.

There is also a temporal shift in seasonal cycles between surface and high altitudes in Switzer-land, as indicated by the difference between urban and mountain sites. Figure 6 compares the four NABEL urban sites with an average altitude of 438.25 m asl with the in situ surface CO obtained at Jungfraujoch with an altitude of 3578 m asl. The low altitude sites located in urban areas show

a similar seasonal variability as the surface CO at Paris, with a maximum around January and a





minimum around July. The surface CO seasonal variation is deeply impacted and driven by local anthropogenic emissions. In comparison, the high altitude NABEL site at Jungfraujoch presents the same seasonal variability of the whole atmosphere (as observed in the total column seasonality) with a time shift of 2 months. The GEOS-Chem monthly averaged surface VMR shows similar variability

as compared to the NABEL surface data at Jungfraujoch. Unlike the Paris case, the GEOS-Chem underestimation for the CO surface VMR is about $-23\%$, which is similar to the difference of $-21\%$ obtained for the total columns, being consistent with much less influence from low altitude emissions and from the PBL.

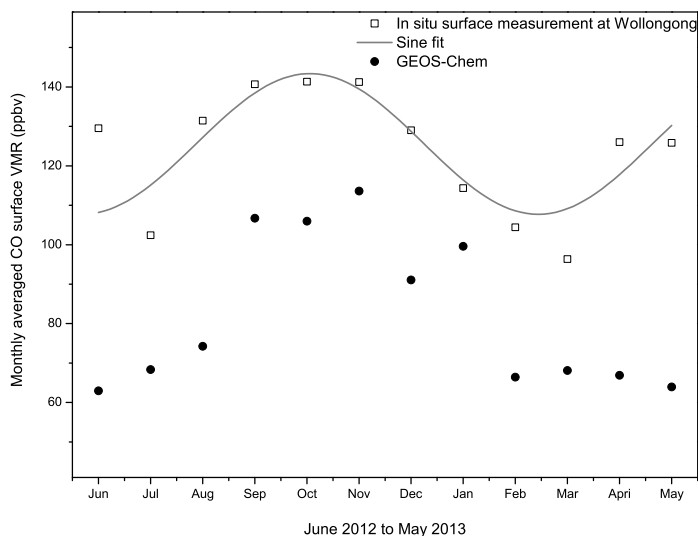

**Figure 7.** Monthly averaged CO surface VMR at Wollongong from in situ measurements (black open squares) and from GEOS-Chem model (black full circles).

Figure 7 shows the monthly averaged surface CO in situ measurements performed at Wollongong

between June 2012 and May 2013. We observe a surface seasonality with a maximum around October and a minimum in February-March. The maximum corresponds to elevated biomass burning levels during the Southern Hemispheric summer (Edwards et al., 2006). This is confirmed by the GEOS-Chem simulation performed without biomass burning emissions, in which the simulated CO seasonality at Wollongong is largely reduced (see Sect. 4.3). The after March increase during the end

of autumn and at the beginning of the winter season corresponds to increased anthropogenic emissions (heating and traffic). The secondary minimum in July can possibly be explained by a reduced influence of local CO emissions from cars on the university campus (where the measurements are




made) during the winter university vacation period. It seems that there is no significant time shift between the CO seasonal variabilities at the Wollongong surface level and at higher altitudes. This

suggests the Wollongong surface atmosphere is generally representative of the free troposphere. The GEOS-Chem monthly averaged surface VMR shows a maximum during the austral spring and a lower level after the end of the austral summer until the austral winter. The background seasonality of CO is mainly driven by biomass burning sources modulated by the OH sink, (Buchholz et al., 2016). Similar to Paris, the surface CO discrepancy between model and measurement of $-33\%$ is

slightly increased as compared to the value of $-20\%$ for the total columns.

### 4.3 Emission sources impacting the seasonality of CO columns

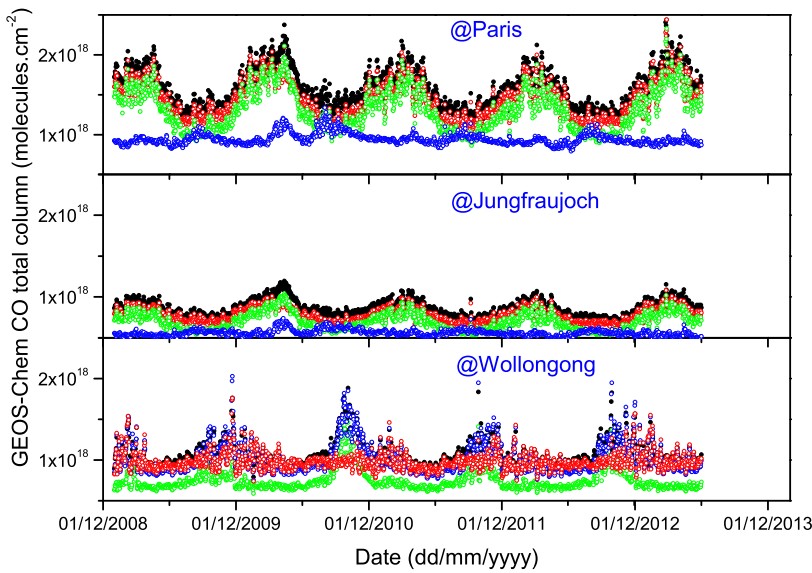

**Figure 8.** GEOS-Chem time series of CO total columns for Paris (top panel), Jungfraujoch (middle panel) and Wollongong (bottom panel). Different colors indicate standard run (black), run without biomass burning (red), run without biogenic emissions (green) and run without anthropogenic emissions (blue).

In order to study the influence of the different categories of CO and NMVOC emissions on the CO total column and its seasonality at the three sites, another three GEOS-Chem simulations were performed. These relied on the same setup as for the standard run (standard chemistry, horizontal

resolution, time period...), but in each of these runs we turned off either the biogenic, the anthropogenic (incorporating the biofuel emissions) or the biomass burning emission sources that are im-





plemented in the model. These categories include direct emissions of CO (for both anthropogenic and biomass burning sources) and of its NMVOC precursors, as well as direct emissions of nitric oxide (NO). In these three sensitivity runs, $CH_4$ concentrations are provided by the NOAA measurements. Hereafter, they are referred to, as the non-biogenic, non-anthropogenic and non-biomass burning simulations. The results from these GEOS-Chem sensitivity simulations are compared to the standard run (results shown in Fig. 3). All four runs cover the mid-2005 to mid-2013 time period, hence starting a few years before the period under investigation here. This allows us to establish a stable situation for the period after 2008 (most of the long-lived precursors of CO are removed from the atmosphere between mid-2005 and 2008). Fig. 8 shows the CO total columns simulated by the different runs of GEOS-Chem for the three sites: the standard run in small black circles, together with the non-biomass burning, non-biogenic and non-anthropogenic runs in red, green and blue circles, respectively. According to the GEOS-Chem simulations, the results for Paris and Jungfraujoch are quite similar. At these two sites, the seasonal variability of the CO loadings is mainly driven by anthropogenic emissions. Indeed, by shutting off the anthropogenic emissions of CO and of its NMVOC precursors, the amplitude of the CO seasonal variation and periodicity are radically reduced. On the other hand, without either biomass burning or biogenic emissions, the seasonal cycle and maximum peaks are only weakly affected as compared to the standard run, with CO columns slightly lower due to missing emissions. At Wollongong, the seasonal variability is mainly influenced by the biomass burning emissions: the highest peaks (e.g. at the end of 2009) disappear when the biomass burning component is removed from the simulation. The biogenic emissions provide more of a large background contribution. Unlike at Paris and Jungfraujoch, anthropogenic emissions are negligible in Wollongong. For the CO surface VMR, the GEOS-Chem sensitivity runs provide the same results as compared to the CO total columns for the three studied sites.

## 5  Conclusions

This paper investigates the seasonal variability of CO total columns at three NDACC and/or TC-CON sites: Paris and Jungfraujoch in the Northern Hemisphere and Wollongong in the Southern Hemisphere. The variability of CO above the PBL has a seasonal maximum in March-April and a minimum in September-October in the Northern Hemisphere. This seasonal cycle is shifted by 6 months in the Southern Hemisphere. For both Northern-hemispheric sites, the seasonal variability of the CO total columns seems to be mainly driven by anthropogenic emissions. On the contrary, the Southern-hemispheric site Wollongong is mainly influenced by the biomass burning contribution. We have compared the ground-based FTIR data to satellite measurements from IASI-MetOp and MOPITT and to GEOS-Chem model outputs, which all of them confirm the observed CO seasonal variability. The GEOS-Chem model also shows that the CO seasonality at Paris and Jungfraujoch is mainly controlled by anthropogenic emissions. This is different to Wollongong, where it is due



to biomass burning. For sites that are strongly affected by local anthropogenic emissions, we have observed a temporal shift in the seasonal patterns at the surface and in the higher atmospheric layers. This is likely because zonal mixing occurs on a shorter (1 – 2 weeks) timescale as compared to com-

plete vertical tropospheric mixing (1 – 2 months). The observed time-lag between upper altitude and surface CO is about 2 months in Paris and at the Jungfraujoch. The 2 months' shift is also confirmed by the GEOS-Chem model. In Wollongong, where low local anthropogenic emissions prevail and which is largely impacted by biomass burning, such a time shift is neither observed nor modelled.

*Acknowledgements.* We are grateful to Université Pierre et Marie Curie and Région Île-de-France for their

financial contributions and to Institut Pierre-Simon Laplace for support and facilities. We thank the National Center for Atmospheric Research MOPITT science team and NASA for producing and archiving the MOPITT CO product. Thanks are also due to the Swiss National Air Pollution Monitoring Network (NABEL) for delivering ground data around Switzerland. The University of Liège contribution to the present work has primarily been supported by the F.R.S. – FNRS, the Fédération Wallonie-Bruxelles and MeteoSwiss (GAW-CH program).

We thank the International Foundation High Altitude Research Stations Jungfraujoch and Gornergrat (HFSJG, Bern). We are grateful to all colleagues who contributed to the acquisition of the FTIR data. The NDACC datasets used here are publicly available from the network database (ftp://ftp.cpc.ncep.noaa.gov/ndacc/station). The Australian Research Council has provided financial support over the years for the NDACC site at Wollongong, most recently as part of project DP110101948. We also acknowledge the important contribution to

the measurement program at Wollongong made by researchers other than those listed as co-authors here, including amongst others, Voltaire Velazco and Nicholas Deutscher. IASI has been developed and built under the responsibility of the French space agency CNES. It is flown onboard the MetOp satellite as part of the Eumetsat Polar System (EPS). The IASI L1 data are received through the Eumetcast near real time data distribution service. IASI L1 and L2 data are stored in the French atmospheric database Ether (http://ether.ipsl.jussieu.fr).

The National Center for Atmospheric Research (NCAR) is sponsored by the National Science Foundation. Any opinions, findings and conclusions or recommendations expressed in the publication are those of the author(s) and do not necessarily reflect the views of the National Science Foundation.

[Table 1 about here.]





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





**Table 1.** Parameters obtained from the sinusoidal fit of the seasonal variability to the CO total columns from ground-based, satellite and GEOS-Chem modeling data

|  | Paris | | Jungfraujoch | | Wollongong | |
|---|---|---|---|---|---|---|
|  | Half-period ($w$) (days) | Amplitude ($A$) (%) | Half-period ($w$) (days) | Amplitude ($A$) (%) | Half-period ($w$) (days) | Amplitude ($A$) (%) |
| Ground-based | $191 \pm 3$ | $14 \pm 1$ | $185 \pm 1$ | $12 \pm 1$ | $178 \pm 1$ | $17 \pm 1$ |
| Satellite data | $183 \pm 1$ | $14 \pm 1$ | $190 \pm 2$ | $12 \pm 1$ | $182 \pm 1$ | $16 \pm 1$ |
| GEOS-Chem | $183 \pm 1$ | $19 \pm 1$ | $182 \pm 1$ | $12 \pm 1$ | $180 \pm 1$ | $13 \pm 1$ |