# Peer review of "Seasonal variability of surface and column carbon monoxide over megacity Paris, high altitude Jungfraujoch and Southern Hemispheric Wollongong stations"

_Atmospheric Chemistry and Physics, 2015_

## Referee Comment (RC1) · Anonymous Referee #1 · 5 Apr 2016

The manuscript 'Seasonal variability of surface and column carbon monoxide over mega-city Paris, high altitude Jungfraujoch and Southern Hemisphere Wollongong stations' by Te et al. examines the seasonal cycles of CO at the surface, in the mid-troposphere and the total column abundances. Measurements at three stations in different environments are compared to retrievals from two satellite-borne instruments and the differences discussed with help from the GEOS-Chem CTM v.9. The manuscript provides a very good compilation of the various data sources and model runs.

General Comments: The topic of this paper is suitable for ACD. The CO distributions above Paris and Wollongong are unique and should be published. Their comparison to multiple independent data is noteworthy. My major concern is whether the analysis convincingly shows a significant difference in the timing of the seasonal cycles in the surface and free troposphere. While I support publication, this issue and other comments below should be addressed beforehand.

Specific comments:

Introduction: This section could be re-written, excluding the first paragraph and the reference to medical studies, instead focusing on topic of the paper: the sources/sinks which determine the atmospheric variability of CO. Previous works should be referenced.

Sections 2 and 3: I found these overly long. It is not clear which information is new or specific for this study, or had been discussed in previous papers. Without removing pertinent information these sections could be written more concisely; and possibly combined. The important details of the instrument measurements with suitable references could be listed in a Table. Any significant differences should be discussed. The key point is to show the three surface FTS instruments are comparable.

P 7, Figure 1: The averaging kernels for all three instruments should be shown as an indication of the comparability of the retrievals.

Figure 2: The curve fits in panels 2 and 3 are hard to see and could be made darker or thicker.

Figures 2 and 7, P 12 L 320: The seasonal variability of the measurements was characterized by a sine function. It provides constant cycle over time. It does not account for variability in the observations and not particularly good at fitting the data. The authors should consider using a function which includes the sine curve while also incorporating the residuals from the curve.

P 9: A figure showing the averaging kernels of the two satellites should also be added.

P 13„ L 365: The GEOS-Chem results tend to be lower than the measurements. The text comments that this is consistent with results from two inverse models. It is likely the forward and inverse models have different reasons for underestimating CO in the NH. The authors should cite results from other forward models.

P 14, Figure 4: The results from the two satellite data sets should be plotted in different colors and fit separately.

P 15, Figure 5, bottom panel: The text states the surface measurements show a maximum in Jan-Feb, however the in situ data are highly variable with a broad maximum from late fall through spring. The FTS and the model PBL results appear to show the seasonal maximum shifted later in the year. Looking at the data and not the smooth curves I find it difficult to identify seasonal offsets among the data. This should be investigated.

Note: The symbol captions on the figure may hide the highest CO in 2008 – 2012.

P 15, L 393-395, and Figure 5 top panel: Why do the IASI partial columns show much larger variability than the Paris FTIR? Why aren't the MOPITT results shown? The discussion beginning line 419 - 'As the lifetime...' is long and confusing. Could it be rewritten simply as: 'In addition to local surface sources, column abundances are influenced by the transport of down wind emission sources.'?

P 16, L 409-411: Does GEOS-Chem account for lower vehicular emissions in Paris during July and Aug? The OH sink is very likely the main factor determining the seasonal minimum. The seasonal and vertical effects of CO oxidation by OH need additional discussion.

Figure 6: This needs revision. The figure caption is wrong. It is not clear if the data points are the mean of multiple years at different sites (the urban sites) or individual years (Jungfraujoch). But they should include an indication of the spread of values.
* * *
Interactive
comment

The red curve doesn't seem to follow the red circles. Does the curve fit account for measurement/aggregation uncertainties? The authors should try a singular value decomposition (accounting for the Y error) or orthogonal distance regression (both Y and X errors).

P 17-18, L. 429-445, Figure 7: This section adds little to the paper. Many previous works have shown the seasonal maximum in the SH reflects biomass burning. The model run without biomass burning emissions shows reduced VMR but a very similar seasonal cycle. One could conclude from Figure 7 that anthropogenic sources contribute equally to biomass burning. I suggest this discussion be removed.

---

## Referee Comment (RC2) · Anonymous Referee #2 · 13 Apr 2016

General Comments

This manuscript presents CO measurements from three ground-based FTIR spectrometers, surface in-situ sensors, and two satellite instruments, along with simulations from the GEOS-Chem model. The seasonal variability of CO total columns, tropospheric columns, and boundary layer mixing ratios is examined. GEOS-Chem simulations are used to identify causes of this season variability and to show how it differs between urban, high-altitude, and Southern Hemisphere sites.

This work presents a new FTIR dataset from Paris and provides an interesting comparison between CO measurements at three very different locations. However, the analysis and interpretation should be significantly revised to strengthen the discussion of the results. The manuscript also has many distracting technical and grammatical errors that need to be corrected. I recommend publication after the issues below are carefully addressed.

Specific Comments

Page 1, line 2 – This sentence ("altitude-dependent seasonal variability") implies that seasonal variability will be examined as a function of altitude, e.g., as vertical profiles. This is not the case, so this sentence should be revised to clarify that the study examines total and partial columns and surface measurements.

Section 1, Introduction – The first paragraph is rather simplistic. This whole section should be revised to provide a more comprehensive review of CO chemistry, sources and sinks including current best quantitative estimates, drivers of seasonal variability, and outstanding questions, to provide a clear motivation for the present work. Don't we already know a lot about CO seasonal variability? What does this work aim to add to current knowledge?

Pages 2-3, lines 47-67 – Inconsistent information is provided for each of the three sites and some material is repeated in Section 2. The paragraph for Paris describes the location and CO trends; the paragraph for Jungfraujoch describes the instrument and air sampled; the paragraph for Wollongong just describes the site. Revise to provide the same information for each site and avoid duplication with information later provided in Section 2.

Sections 2.1, 2.2, 2.3 – These more detailed instrument descriptions are also inconsistent with regard to the information provided. Provide the same information about all three FTIR instruments. Some of this information could also be summarized more efficiently in a table.

Sections 3.1, 3.2, 3.3 – Similarly, review these three sections to ensure the same level of detail is provided for each data set. e.g., DOFS, source of a priori data, etc. What terms are included in the random uncertainty error estimates? Are they the same terms for all three sites? What are the systematic errors?

Page 7, Figure 1 – Include averaging kernels for all three sites and comment on similarities and differences. Add total column averaging kernels and sensitivity. Could show the seasonal average AVKs used for smoothing (as described on page 11, line 290). Also plot the AVKs for IASI and MOPITT. Page 9 – give DOFS for IASI and MOPITT.

Page 7, line 198 and page 9, lines 219 and 222 – WACCM v6 is the current version being used by the NDACC IRWG. Is v4 correct?

Page 10, lines 295-305 – Why are the in-situ data daily for Paris, but monthly averages for Wollongong and Jungfraujoch? Are the Paris data used to generate monthly averages too? Clarify.

Page 12, line 318 – Is a simple sine function the most appropriate function to use to fit the time series? Explain in the text.

Page 12, lines 327 – Is 184 +- 4 days the period for some combination of Paris and Jungfraujoch data? Why is no uncertainty given for 191 days? Line 335 – what is the period for Wollongong?

Page 12, lines 347-349 – Was the MOPITT vertical resolution really increased by interpolating between the pressure levels? Correct this sentence.

Page 13, line 365 – Did these inverse modeling studies use GEOS-Chem?

Page 14, Figure 4 – Give the correlation coefficient $R^2$, as was done in Figure 3. Why combine the data from the two satellite instruments in these correlation plots?

Page 14, line 369 – Since GEOS-Chem, and presumably other models, capture the seasonal variability of CO, this implies that we have good knowledge of the processes

controlling it. So what new knowledge is this study contributing? This needs to be better articulated in the Introduction and Conclusions.

Page 14, lines 380-381 – Why exclude the data influenced by local processes, since the study is evaluating causes of variability? Line 384 then says that local CO emissions still affect the variability. This should be better explained.

Page 15, Figure 5 – This caption is incorrect (only shows Paris) and provides insufficient information about what is shown.

Page 15, line 390 – It is difficult to do a meaningful comparison of seasonal cycles just by inspection of Figures 2 and 5.

Pages 15 – There is insufficient discussion of the role of sunlight and OH oxidation on the seasonal cycle of CO.

Page 15, especially lines 400-405 – Various statements about attribution are made in this paragraph but without a clear justification, e.g., "At Paris, the seasonality introduced by these distant sources outweighs the contribution of the local surface." This paragraph should be strengthened. Same comment for lines 421-422 – what is the basis for the statement "The surface CO seasonal variation is deeply impacted and driven by local anthropogenic emissions."? No back-trajectory, Lagrangian, or adjoint modelling is done to back up statements about transported sources. A stronger case needs to be made for all attribution statements.

Page 16, Figure 6 – This caption is completely incorrect.

Pages 18-19 – The discussion of the relative importance of different sources to the seasonal variability of CO at the three sites should also be strengthened. Source strengths should be compared to the underlying seasonal cycle due to oxidation of CO by OH, which ultimately depends on sunlight. Results should be put in context with references to the literature, including some of those cited in the manuscript. A few other possible examples: Derwent et al., Obs and interpretation of seasonal cycles of . . .

ozone and carbon monoxide ... Atmos. Env. 1998. Holloway et al., Global distribution of carbon monoxide, JGR, 2000. Duncan et al., Global budget of CO, 1988–1997: Source estimates and validation with a global model, JGR 2007. Zellweger et al., Intercomparison of four different carbon monoxide measurement techniques and evaluation of the long-term carbon monoxide time series of Jungfraujoch, ACP 2009. etc.

Conclusions – State clearly what new information this work contributes to our understanding of atmospheric CO abundance and variability.

Technical Corrections

Page 1, line 1 – a key atmospheric species

Page 1, line 2 – altitude-dependent

Page 1, line 3 – at three different sites: Paris

Page 1, line 9 – by the IASA-MetOp

Page 1, line 9 – define FTIR

Page 1, line 12 – near-surface

Page 1, line 14 – identification

Page 1, line 15 – delete "on top of"

Page 2, line 18 – Revise "between the surface and above the boundary layer" – between the surface and the free troposphere?

Page 2, line 21 – delete "of"

Page 2, line 30 – defective

Page 2, line 33 – energy-related

Page 2, line 35 – CO

Page 2, line 42 – high-resolution

Page 2, line 43 – define FTIR here, not on line 68

Page 3, line 50 – has been continuously

Page 3, line 70 – and have monitored the

Page 4, line 85 – FTS-Paris is a model

Page 4, line 95 – signal-to-noise ratio

Page 4, line 96 – detector provides coverage of the spectral

Page 4, line 110 – delete "network"

Page 6, line 161 – were recorded over 3 min intervals at the

Page 6, line 162 – were analyzed.

Page 6, line 169 – using the HITRAN 2008 . . . delete "as"

Page 6, line 178 – Figure 1 shows

Page 7, line 187 – in Dils et al. (2011).

Page 8, line 202 – down to the

Page 8, line 209 – Barrett et al. (2003),

Page 8, lines 202, 210, 211 – inconsistent dashes for number ranges

Page 9, line 221 – signal-to-noise ratio

Page 9, line 230 – delete "from the IASI sounder on the MetOp satellite"

Page 9, line 238 – profiles

Page 9, line 242 – using version 6 retrievals

Page 10, line 252 – can be used to simulate global

[Figure]

**[ACPD](ACPD)**

Interactive
comment

Page 10, lines 251-252 – (Bey et al., 2011; Park et al., 2004; etc.)

Page 11, line 312 – regular basis

Page 11, line 316-317 – delete "the low altitude . . . here."

Page 12, line 339 – delete "satellite"

Page 12, line 343 – high-altitude

Page 12, line 345 – footprint not only includes the site, but

Page 13, Figure 3 caption – ground-based. Also describe panels in order from top to bottom, not top, middle, bottom.

Page 13, line 354 – Text says satellite data for Jungfraujoch is shown in Figure 3, but so are data for Paris and Wollongong.

Page 14, line 384 – due to anthropogenic

Page 15, line 401 – what is meant by "warming system"?

Page 16, line 410 – at the end of

Page 16, line 411 – at the end of

Page 16, line 419 – low-altitude sites

Page 16, line 420 – variability to the

Page 17, line 434 – The increases after March

Page 19, line 484 – which confirm

---

## Author Comment (AC1) · 15 Jun 2016

Ref.: ACP-2015-884

We would like to thank the both anonymous referees for their reviews and very constructive remarks. These were taken into account the new version of the paper (online revised paper: "CO_Variability2.pdf").

In the following text, we answer to the individual comments. As many modifications were added, we join as supplement the "CO_Variability-difference.pdf" file with "Revisions" mode to track all changes.

Best regards, Y. Té

Referee#1 Specific Comments

Comment#01

Introduction: This section could be re-written, excluding the first paragraph and the reference to medical studies, instead focusing on topic of the paper: the sources/sinks which determine the atmospheric variability of CO. Previous works should be referenced.

Answer#01

- As proposed, we have re-written the "Introduction" section. The previous first paragraph has been removed as well as references related to medical studies.

- We have added some sentences with references concerning studies of sources and sinks of CO.

- We have homogenized the description of the three sites (some information concerning the Paris site were removed).

- The description of the different sections is also updated.

Comment#02

Sections 2 and 3: I found these overly long. It is not clear which information is new or specific for this study, or had been discussed in previous papers. Without removing pertinent information these sections could be written more concisely; and possibly combined. The important details of the instrument measurements with suitable references could be listed in a Table. Any significant differences should be discussed. The key point is to show the three surface FTS instruments are comparable.

Answer#02

- As recommended by both referees, we have added a new Table, which lists relevant

information concerning the ground-based FTIR instruments and the optical configuration used to record data exploited by the present paper. This Table contributes to shortening the two sections.

- The new Table clearly shows that the three ground-based FTIR instruments are comparable.

- We have also merged information on the measurement sites (previous section 2) and on the data (previous section 3) in order to construct 2 new sections: the new section 2 is concerned the description of the remote sensing instruments and data (ground-based and satellite) and the new section 3 with the in situ measurements and the GEOS-Chem modeling data.

- They are much more streamlined than in the previous version.

Comment#03

P 7, Figure 1: The averaging kernels for all three instruments should be shown as an indication of the comparability of the retrievals.

Answer#03

- The averaging kernels (AVKs) for Paris have been shown in order to justify the use of two different partial columns, one for the boundary layer and the other for the free troposphere. Since we don't use such data at the other two sites (Jungfraujoch and Wollongong), we don't see the need of showing their AVKs which can be found for example in Barret et al. (2003) and Buchholz (PhD thesis, http://ro.uow.edu.au/theses/4272/). Even more so as we are not concerned with a technical paper where we propose a new method to improve the data retrieval. We agree that such a presentation would need to be included if we were to make a complete instrument comparison of the three sites, which would probably be the topic for publication in AMT rather than ACP.

- In order to provide consistent information for each site, we have moved the figure into section 4.2 (figure 1 moved to figure 5), where the information is really exploited.

- The following sentences are added in Section 4: "As mentioned in Section 2.2, the retrieval grid (49 levels) corresponds to a much thinner atmospheric layering than the effective vertical resolution indicated by the averaging kernels (Rodgers, 1990). The CO averaging kernels for each altitude of the a priori profile indicate a good sensitivity of the FTS-Paris instrument to the PBL. Indeed, Figure 5 shows that the retrieval of CO essentially provides two independent measurement points of CO in the troposphere: the first point supplies maximal information in the altitude range between 0 and 1000 m and thus well represents the PBL. The second one is representative of the upper troposphere, with a maximum around 8-9 km."

Comment#04

Figure 2: The curve fits in panels 2 and 3 are hard to see and could be made darker or thicker.

Answer#04

- Done, we now use the darker color "dark gray" (Figure 2 is now Figure 1).

Comment#05

Figures 2 and 7, P 12 L 320: The seasonal variability of the measurements was characterized by a sine function. It provides constant cycle over time. It does not account for variability in the observations and not particularly good at fitting the data. The authors should consider using a function which includes the sine curve while also incorporating the residuals from the curve

Answer#05

- We have added the residuals of the sine function fit in Figure 1 (old Figure 2), so that the reader can assess the quality of the fit. The merit of the simple sine function is that it provides only few key parameters that can be compared in quite different situations (e.g. remote and polluted sites). One could wonder whether neglecting the signal complexity in the urban atmosphere would change any of the conclusions

(considering that the remote Jungfraujoch data seem to be well reproduced by the sine function). But the surface-column comparison between Wollongong and Paris shows that an atmosphere with pollution spikes does not necessarily lead to a shift when one compares with the background data. We have therefore renounced from adding more complexity to the fit function.

Comment#06

P 9: A figure showing the averaging kernels of the two satellites should also be added.

Answer#06

- Please see above (discussion referring to remark on P7, Figure 1) for our explanation concerning the AVKs of the ground based FTIRs.

- We have added the AVKs of IASI-MetOp around the Paris site (cf. figure 5) in order to motivate why in Fig.4 we only calculate the free tropospheric column for IASI-MetOp between 2 and 12 km and neglect the surface. The summer AVKs of IASI-MetOp (but also the winter ones not shown in the paper) show less surface sensitivity. We therefore only exploit the partial columns corresponding to the free troposphere.

- For MOPITT, we prefer to refrain from showing its AVKs, because we only use CO total columns and its AVKs are already published elsewhere (Worden et al. (2010), Jiang et al. (2013, doi:10.1002/jgrd.50216), Buchholz PhD thesis (see above)).

- In agreement with requests from the second reviewer we add more information on the satellite DOF by citing the original literature: George et al. (2009) for IASI and Worden et al 2010 for MOPITT in section 2.4 on the satellite instruments. The DOF are about 2 for both instruments.

Comment#07

P 13" L 365: The GEOS-Chem results tend to be lower than the measurements. The text comments that this is consistent with results from two inverse models. It is likely

the forward and inverse models have different reasons for underestimating CO in the NH. The authors should cite results from other forward models.

Answer#07

- We provide some additional references that compare observations and forward models. These confirm our observation: "In Duncan et al. (2007), the averaged bias between observations and GEOS-Chem model simulations is less than +- 10%, but for some sites (Seychelles or Tae-Ahn), the bias can exceed +- 20%. Zeng et al. (2015) have observed large underestimations from models as compared to the ground-based FTIR stations in the Southern Hemisphere from -19.2% to -27.5% depending on the emission inventories implemented in the models for the Wollongong site (episodic events unaccounted for in the emission inventories)."

Comment#08

P 14, Figure 4: The results from the two satellite data sets should be plotted in different colors and fit separately.

Answer#08

- Done. We have added the results obtained by the robust linear fit applied on the IASI-MetOp data and on the MOPITT data taken separately. The robust fitting technique is explained in the text.

Comment#09

P 15, Figure 5, bottom panel: The text states the surface measurements show a maximum in Jan-Feb, however the in situ data are highly variable with a broad maximum from late fall through spring. The FTS and the model PBL results appear to show the seasonal maximum shifted later in the year. Looking at the data and not the smooth curves I find it difficult to identify seasonal offsets among the data. This should be investigated.

Answer#09

- We have modified the previous figure 5 (now Figure 4) and just show monthly data as five year averages for the surface in situ measurement at the Paris site, like those from the other two sites (because we have only monthly averaged data for Switzerland (cf. Figure 6) and for Wollongong, we have monthly averaged the in situ data (cf. Figure 7)). This approach is also more consistent with the remote sensing data where the sine fit has been extended over several years.

- This makes the time shift much more evident.

Comment#10

Note: The symbol captions on the figure may hide the highest CO in 2008 – 2012.

Answer#10

- This was corrected. Highest CO values are no more hidden in the new figure 4 displaying now monthly averages.

Comment#11

P 15, L 393-395, and Figure 5 top panel: Why do the IASI partial columns show much larger variability than the Paris FTIR? Why aren't the MOPITT results shown? The discussion beginning line 419 - 'As the lifetime...' is long and confusing. Could it be rewritten simply as: 'In addition to local surface sources, column abundances are influenced by the transport of down wind emission sources.'?

Answer#11

- The IASI-MetOp data don't show a "larger variability", for the "free tropospheric" partial columns as compared to the total columns (similar). Depending on the thermal contrast, the IASI-MetOp might have less sensitivity to the surface signal, cf. George et al., 2009. This may explain the same variability between total and partial columns. For the FTS-Paris data, the difference is due to the partial columns which get narrower

than the total columns. Because a significant contribution of points with large variations is coming from the PBL contribution. And this large contribution is subtracted from the free troposphere column.

- We don't have any additional partial columns of MOPITT around the Paris site (in the contrary to IASI-MetOp). Worden et al. (2010) noticed an increase of the DOFs (limited to 2) by using TIR and NIR retrievals. But we don't know the relevancy of the free troposphere partial columns. In doubt, we prefer to avoid using them.

- We have also rewritten the sentence as proposed.

Comment#12

P 16, L 409-411: Does GEOS-Chem account for lower vehicular emissions in Paris during July and Aug? The OH sink is very likely the main factor determining the seasonal minimum. The seasonal and vertical effects of CO oxidation by OH need additional discussion.

Answer#12

- Low vehicle traffic is likely included in GEOS-Chem, but still GEOS-Chem underestimates the observations. We agree that the minimum in July-August is mainly due to increased OH in summer and this has been lacking in our discussion. We have thus modified the text accordingly: "The minimum in July-August, where the PBL height is highest, not only corresponds to increased oxidization of CO by OH, whose abundance is influenced by solar ultraviolet radiation (Bousquet et al., 2005; Rohrer and Berresheim, 2006; Duncan et al., 2007), but also to the summer vacation season during which the inhabitants of Paris usually leave the city (by more than 50%, http://www.insee.fr), leading to a drastic decrease of vehicle traffic."

Comment#13

Figure 6: This needs revision. The figure caption is wrong. It is not clear if the data points are the mean of multiple years at different sites (the urban sites) or individual

years (Jungfraujoch). But they should include an indication of the spread of values. The red curve doesn't seem to follow the red circles. Does the curve fit account for measurement/aggregation uncertainties? The authors should try a singular value decomposition (accounting for the Y error) or orthogonal distance regression (both Y and X errors).

Answer#13

- The caption has been corrected.

- We have improved the clarity of Figure 6 and only show the 5-year averages.

- From the new figure it is clear that the fit follows the data and there is no need to exploit other fitting techniques.

- We have added the residuals of the sine function fit in Figure 7 in order to show the consistency of our sine function fit with the data.

Comment#14

P 17-18, L. 429-445, Figure 7: This section adds little to the paper. Many previous works have shown the seasonal maximum in the SH reflects biomass burning. The model run without biomass burning emissions shows reduced VMR but a very similar seasonal cycle. One could conclude from Figure 7 that anthropogenic sources contribute equally to biomass burning. I suggest this discussion be removed.

Answer#14

- In order to avoid redundant information, we have removed the following sentences: "This is confirmed by the GEOS-Chem simulation performed without biomass burning emissions, in which the simulated CO seasonality at Wollongong is largely reduced. The secondary minimum in July can possibly be explained by a reduced influence of local CO emissions from cars on the university campus (where the measurements are made) during the winter university vacation period."

- However, we have kept the information concerning the "increase after March", which is observed by observations and not by GEOS-Chem. This adds complementary and relevant information, showing that some episodic emission sources are not accounted for in the emission inventories implemented in GEOS-Chem. In order to better understand this issue, we would need measurements over longer period of time.

- Concerning the reviewers statement "One could conclude from figure 7 that anthropogenic sources contribute equally to biomass burning", we would like to point out that apart from that he or she probably meant Fig. 8 instead of Fig. 7, we don't quite agree with this interpretation. If we look at the bottom panel (for Wollongong site), we can see that turning off the biomass burning (in red) has a large impact on some broad "seasonal" peaks and that removing biogenic emissions (in green) has a large but non-seasonal effect, whereas removal of anthropogenic effects (in blue) in the modeling has only a very small impact. In consequence, no such modification was made.

Referee#2

Specific Comments

Comment#01

Page 1, line 2 – This sentence ("altitude-dependent seasonal variability") implies that seasonal variability will be examined as a function of altitude, e.g., as vertical profiles. This is not the case, so this sentence should be revised to clarify that the study examines total and partial columns and surface measurements.

Answer#01

- We agree and we have modified the sentence: "The paper studies the seasonal variation of surface and column CO at three different sites (Paris, Jungfraujoch and Wollongong), . . .".

Comment#02

[Figure]

Section 1, Introduction – The first paragraph is rather simplistic. This whole section should be revised to provide a more comprehensive review of CO chemistry, sources and sinks including current best quantitative estimates, drivers of seasonal variability, and outstanding questions, to provide a clear motivation for the present work. Don't we already know a lot about CO seasonal variability? What does this work aim to add to current knowledge?

Answer#02

- We have re-written the "Introduction" section. The previous first paragraph was removed as well as the references related to medical studies.

- We have included references related to studies of sources and sinks of CO.

- We have homogenized the description of the three sites (some information concerning the Paris site were removed).

- The description of the different sections is also updated.

Comment#03

Pages 2-3, lines 47-67 – Inconsistent information is provided for each of the three sites and some material is repeated in Section 2. The paragraph for Paris describes the location and CO trends; the paragraph for Jungfraujoch describes the instrument and air sampled; the paragraph for Wollongong just describes the site. Revise to provide the same information for each site and avoid duplication with information later provided in Section 2.

Answer#03

- The criticism has also been brought up by the first referee. Here we repeat how we have answered the criticism before. As recommended by both referees, we have added a new Table, which lists relevant information concerning the ground-based FTIR instruments and the optical configuration used to record data exploited by the present

paper. This Table contributes to shortening the two sections.

- The new Table clearly shows that the three ground-based FTIR instruments are comparable (in technical and functional terms). We have also merged information on the measurement sites (previous section 2) and on the data (previous section 3) in order to construct 2 new sections: the new section 2 concerns the description of the remote sensing instruments and data (ground-based and satellite) and the new section 3 concerns the in situ measurements and the GEOS-Chem modeling data.

Comment#04

Sections 2.1, 2.2, 2.3 – These more detailed instrument descriptions are also inconsistent with regard to the information provided. Provide the same information about all three FTIR instruments. Some of this information could also be summarized more efficiently in a table.

Answer#04

- This has been done, please see above.

Comment#05

Sections 3.1, 3.2, 3.3 – Similarly, review these three sections to ensure the same level of detail is provided for each data set. e.g., DOFS, source of a priori data, etc. What terms are included in the random uncertainty error estimates? Are they the same terms for all three sites? What are the systematic errors?

Answer#05

- We have tried to gather the same type of information concerning the retrieval of the ground-based FTIR data for each of the sites.

- We have added the DOF for each ground-based FTIR and for the satellite instruments.

- Random errors: terms considered as random errors are temperature profile, instrument noise, solar zenith angle, interfering solar lines. These details can be found in the references mentioned in the text: "These sources have been investigated in detail by Té et al. (2012), following the procedure outlined by Rinsland et al. (2000). According to this evaluation, the random uncertainty is around 2.5%."

- Systematic errors: We have also added a sentence which discusses systematic errors: "Concerning the systematic uncertainties of about 3 to 6.8% (Té et al., 2012), the largest source is linked to the quality of available spectroscopic parameters (line intensity and air broadened half-width uncertainties in Rothman et al. (2009), which is similar for the three sites".

Comment#06

Page 7, Figure 1 – Include averaging kernels for all three sites and comment on similarities and differences. Add total column averaging kernels and sensitivity. Could show the seasonal average AVKs used for smoothing (as described on page 11, line 290). Also plot the AVKs for IASI and MOPITT.

Answer#06

- Similar requests have been made by the first referee. Here we repeat our previous arguments and give some new information.

- The averaging kernels (AVKs) for Paris have been shown in order to justify the use of two different partial columns, one for the boundary layer and the other for the free troposphere. Since we don't use such data at the other two sites (Jungfraujoch and Wollongong), we don't see the need of showing their AVKs which can be found for example in Barret et al. (2003) and Buchholz (PhD thesis, http://ro.uow.edu.au/theses/4272/). Even more so as we are not concerned with a technical paper where we propose a new method to improve the data retrieval. We agree that such a presentation would need to be included if we were to make a complete instrument comparison of the three sites, which would probably be the topic for publication in AMT rather than ACP.

- In order to provide consistent information for each site, we have moved the figure into section 4.2 (figure 1 moved to figure 5), where the information is really exploited. In this way, we have added the AVKs of IASI-MetOp around Paris, in order to motivate why in Fig. 4 we only calculate the free tropospheric column for IASI-MetOp between 2 and 12 km and neglect the surface. The summer AVKs of IASI-MetOp (but also the winter ones not shown in the paper) show less surface sensitivity. We therefore only exploit the partial columns corresponding to the free troposphere.

- For MOPITT, we prefer to refrain from showing its AVKs, because we only use CO total columns and its AVKs are already published elsewhere (Worden et al. (2010), Jiang et al. (2013, doi:10.1002/jgrd.50216), Buchholz PhD thesis (see above)).

- We also prefer not showing the AVKs of FTIR instruments from Jungfraujoch and from Wollongong, because we don't use them in our discussions. Moreover, the smoothing did not change the comparison results between the model and our observations (difference smaller than 1%).

Comment#07

Page 9 – give DOFS for IASI and MOPITT.

Answer#07

- We have added the requested information on the satellite DOF by citing the original literature: George et al. (2009) for IASI and Worden et al 2010 for MOPITT in section 2.4 on the satellite instruments. The DOF are about 2 for both of the instruments.

Comment#08

Page 7, line 198 and page 9, lines 219 and 222 – WACCM v6 is the current version being used by the NDACC IRWG. Is v4 correct?

Answer#08

- Version 4 is still used for the Jungfraujoch and Wollongong retrievals. The switch

to version 6 will happen with the next release of the data set. However, we do not anticipate a significant impact on the retrieval.

Comment#09

Page 10, lines 295-305 – Why are the in-situ data daily for Paris, but monthly averages for Wollongong and Jungfraujoch? Are the Paris data used to generate monthly averages too? Clarify.

Answer#09

- Daily surface in situ measurements are available for the Paris site. But for the other two sites (Jungfraujoch and Wollongong), only monthly averages were available.

- In order to improve the consistency of our comparison, we present the Paris data also as monthly averages. Figure 4 has been updated correspondingly.

Comment#10

Page 12, line 318 – Is a simple sine function the most appropriate function to use to fit the time series? Explain in the text.

Answer#10

- Surely not, but this simple periodic function has been largely used in previous studies of seasonal variations. We have motivated our choice by adding the following phrase: "We have used a sine function Eq. (1) to characterize this seasonal variability in agreement with previous studies conducted by Rinsland et al. (2000); Rinsland et al. (2001); Rinsland et al. (2007) and by Zhao et al. (2002). In comparison to Rinsland et al., we have removed the linear term, because our data sets do not show any significant trend."

Comment#11

Page 12, lines 327 – Is 184 +- 4 days the period for some combination of Paris and

Jungfraujoch data? Why is no uncertainty given for 191 days? Line 335 – what is the
period for Wollongong?

Answer#11

- The 184 +- 4 days corresponds to the average of all three sites.

- We have re-calculated the value for the "Northern Hemisphere", taking only Paris
Jungfraujoch data into account. We now write: "The average amplitude of the seasonal
variability is about (13 +- 3)% of the column average and the average half-cycle is about
188 +- 4 days for the Northern hemispheric sites."

- We have added the standard deviation for 191 days => 191 +- 3 days.

- The period for Wollongong has also been included in the text (178 +- 1 days).

Comment#12

Page 12, lines 347-349 – Was the MOPITT vertical resolution really increased by inter-
polating between the pressure levels? Correct this sentence.

Answer#12

- Probably we haven't been clear enough in the original manuscript. Certainly, the
interpolation does not change (nor improve) the vertical resolution of MOPITT. We just
wanted to explain how we interpolated the data in order to extract the total column from
a given altitude to the top of the atmosphere. We have changed the sentence: "For the
MOPITT data, we extracted the retrieved CO profile for Jungfraujoch and interpolated
the lower pressure levels on a thinner vertical grid in order to calculate the column
between the given ground altitude and the top of the atmosphere."

Comment#13

Page 13, line 365 – Did these inverse modeling studies use GEOS-Chem?

Answer#13

- Kopacz et al, have used GEOS-Chem and Hooghiemstra et al., used the TM5-CO model.

Comment#14

Page 14, Figure 4 – Give the correlation coefficient RЁĘ2, as was done in Figure 3. Why combine the data from the two satellite instruments in these correlation plots?

Answer#14

- We have included Rˆ2.

- We have added the results obtained by the robust linear fit applied on the IASI-MetOp data and on the MOPITT data taken separately. This doesn't change anything.

Comment#15

Page 14, line 369 – Since GEOS-Chem, and presumably other models, capture the seasonal variability of CO, this implies that we have good knowledge of the processes controlling it. So what new knowledge is this study contributing? This needs to be better articulated in the Introduction and Conclusions.

Answer#15

- We have used different datasets (using different techniques) to present first evidence of a 2 months time-lag between surface and column seasonal variabilities under specific conditions (local urban anthropogenic emission sources) for Paris and Jungfraujoch sites. Indeed, GEOS-Chem captures the seasonality of CO (both at the surface and for the column) for 3 sites with specific location characteristics and the time-lag of 2 months. Future and more detailed studies including more sites might be undertaken to further analyse the time lag parameter (magnitude on a global scale, influence of urbanization level of the sites (local and non-local emission sources), . . .).

- We have strongly modified the "Introduction" section (see above modifications "Section 1, Introduction") and we better explain that we investigate differences in surface

and upper tropospheric CO using different remote sensing methods in combination with surface analyzers. We also have newly structured the "Conclusion" section to better explain the significance of our paper. As all other changes made in the original manuscript, these have been highlighted.

Comment#16

Page 14, lines 380-381 – Why exclude the data influenced by local processes, since the study is evaluating causes of variability? Line 384 then says that local CO emissions still affect the variability. This should be better explained.

Answer#16

- We have removed the ambiguous phrase (380-381), because we now show all data. There is nothing excluded anymore. We hope that this removes all confusion.

Comment#17

Page 15, Figure 5 – This caption is incorrect (only shows Paris) and provides insufficient information about what is shown.

Answer#17

- The caption was wrong and this has been corrected.

- The caption of the figure (number 4 in the new manuscript version) now reads: "Free tropospheric (top) and surface (bottom) CO at Paris as monthly averages over 5 years. CO VMR in the PBL come from the in situ CO analyser (dark green hexagons) and from the FTS-Paris (magenta stars), as well as from the GEOS-Chem model (black circles). Free tropospheric CO columns were calculated between 2 and 12 km and monthly averaged over the period from 2009 to 2013. Shown are data from IASI-MetOp (orange squares), FTS-Paris (purple stars) and from GEOS-Chem modeling (black circles). A sine function fit is applied to the IASI-MetOp and in situ CO data."

Comment#18

Page 15, line 390 – It is difficult to do a meaningful comparison of seasonal cycles just by inspection of Figures 2 and 5.

Answer#18

- We have modified Figure 5. We think that the comparison between the figure 1 (2 previously) and Figure 4 (previously 5) is much clearer in the revised manuscript version.

Comment#19

Pages 15 – There is insufficient discussion of the role of sunlight and OH oxidation on the seasonal cycle of CO.

Answer#19

- We have corrected for this and have added the following: "The minimum in July-August, where the PBL height is highest, not only corresponds to an increased ox-idization of CO by OH, whose abundance is influenced by solar ultraviolet radiation (Bousquet et al., 2005; Rohrer et al., 2006; Duncan et al., 2007), but also to the sum-mer vacation season during which the inhabitants of Paris usually leave the city (by more than 50%, http://www.insee.fr), leading to a drastic decrease of vehicle traffic."

Comment#20

Page 15, especially lines 400-405 – Various statements about attribution are made in this paragraph but without a clear justification, e.g., "At Paris, the seasonality intro-duced by these distant sources outweighs the contribution of the local surface." This paragraph should be strengthened. Same comment for lines 421-422 – what is the basis for the statement "The surface CO seasonal variation is deeply impacted and driven by local anthropogenic emissions."? No back-trajectory, Lagrangian, or adjoint modelling is done to back up statements about transported sources. A stronger case needs to be made for all attribution statements.

Answer#20

- Since we do not intend to add additional modeling (back trajectories, etc.) the incriminated phrases have been removed.

Comment#21

Page 16, Figure 6 – This caption is completely incorrect.

Answer#21

- This has been corrected. The caption now reads: "5 years monthly averaged CO in situ measurements at the surface in Switzerland using Swiss NABEL data from 2009 to 2013 (dark squares for urban sites and red diamonds for Jungfraujoch) The sine function fit is applied on the 5 years monthly means of urban sites (black lines) and of mountain site (red line). Monthly averaged CO surface VMR from GEOS-Chem model located at Jungfraujoch (black full circles)."

Comment#22

Pages 18-19 – The discussion of the relative importance of different sources to the seasonal variability of CO at the three sites should also be strengthened. Source strengths should be compared to the underlying seasonal cycle due to oxidation of CO by OH, which ultimately depends on sunlight. Results should be put in context with references to the literature, including some of those cited in the manuscript. A few other possible examples: Derwent et al., Obs and interpretation of seasonal cycles of ozone and carbon monoxide ... Atmos. Env. 1998. Holloway et al., Global distribution of carbon monoxide, JGR, 2000. Duncan et al., Global budget of CO, 1988–1997: Source estimates and validation with a global model, JGR 2007. Zellweger et al., Inter-comparison of four different carbon monoxide measurement techniques and evaluation of the long-term carbon monoxide time series of Jungfraujoch, ACP 2009. etc.

Answer#22

- We have extended the discussion by including some of the proposed references and discussion of the role of CO+OH has been included (see above).

- In the discussion we added "This is in agreement with previous model studies. Duncan et al. (2007) show that fossil fuel emissions are the main contribution to the CO burden in the Northern extra-tropics. The high-altitude Jungfraujoch site is strongly impacted by long-range transport of CO. At least one third of it is of non-European origin origin (Duncan et al., 2007; Zellweger et al., 2009)." And "Zeng et al. have observed that the impact of biogenic emissions on CO is larger in the Southern Hemisphere than in the Northern."

Comment#23

Conclusions – State clearly what new information this work contributes to our understanding of atmospheric CO abundance and variability.

Answer#23

- We have modified the "Conclusion" section in order to point out more clearly the time-lag of 2 months between surface and column CO for urban sites. As far as we know, there is no study which has mentioned this time-lag.

- This paper is the first step in the understanding/characterisation of this time-lag. Further studies might be undertaken by including more sites in order to check the dependence of this time-lag value on the latitude for example, or on the nature and strengths of the sources.

Technical Corrections

Page 1, line 1 – a key atmospheric species

- Done.

Page 1, line 2 – altitude-dependent

- Does not apply any more as we have changed the text.

Page 1, line 3 – at three different sites: Paris

- Done.

Page 1, line 9 – by the IASA-MetOp

- Done.

Page 1, line 9 – define FTIR

- Done.

Page 1, line 12 – near-surface

- Done.

Page 1, line 14 – identification

- Done.

Page 1, line 15 – delete "on top of"

- Replaced by "at".

Page 2, line 18 – Revise "between the surface and above the boundary layer" – between the surface and the free troposphere?

- We have modified the sentence to "between surface measurements and total column data".

Page 2, line 21 – delete "of"

- Done.

Page 2, line 30 – defective

- Done.

Page 2, line 33 – energy-related

- Done.

Page 2, line 35 – CO

- Done.

Page 2, line 42 – high-resolution

- Done.

Page 2, line 43 – define FTIR here, not on line 68

- Done, FTIR is defined on line 8.

Page 3, line 50 – has been continuously

- Done.

Page 3, line 70 – and have monitored the

- Done.

Page 4, line 85 – FTS-Paris is a model

- The sentence has been changed to "FTS-Paris is a Michelson interferometer from Bruker Optics".

Page 4, line 95 – signal-to-noise ratio

- Done.

Page 4, line 96 – detector provides coverage of the spectral

- Done.

Page 4, line 110 – delete "network"

- Done.

Page 6, line 161 – were recorded over 3 min intervals at the

- The sentence is removed, information in the newly added Table 1.

Page 6, line 162 – were analyzed.

- We have used "were analysed".

Page 6, line 169 – using the HITRAN 2008 . . . delete "as"

- Done.

Page 6, line 178 – Figure 1 shows

- Done.

Page 7, line 187 – in Dils et al. (2011).

- Done.

Page 8, line 202 – down to the

- Done.

Page 8, line 209 – Barrett et al. (2003),

- We have modified to "Barret et al. (2003)".

Page 8, lines 202, 210, 211 – inconsistent dashes for number ranges

- Done.

Page 9, line 221 – signal-to-noise ratio

- Done.

Page 9, line 230 – delete "from the IASI sounder on the MetOp satellite"

- Done.

Page 9, line 238 – profiles

- Done.

Page 9, line 242 – using version 6 retrievals

- Done.

Page 10, line 252 – can be used to simulate global

- Done.

Page 10, lines 251-252 – (Bey et al., 2011; Park et al., 2004; etc.)

- Done.

Page 11, line 312 – regular basis

- Done.

Page 11, line 316-317 – delete "the low altitude . . . here."

- Done.

Page 12, line 339 – delete "satellite"

- Done.

Page 12, line 343 – high-altitude

- Done.

Page 12, line 345 – footprint not only includes the site, but

- Done.

Page 13, Figure 3 caption – ground-based. Also describe panels in order from top to bottom, not top, middle, bottom.

- Done. The new caption is "Time series of CO columns from satellite instruments and ground-based FTIR are given for Paris, Jungfraujoch and Wollongong (from top to

bottom). IASI-MetOp and MOPITT columns are displayed as black open squares and blue open circles. Ground-based FTIR CO total columns are shown by red stars, green triangles and cyan diamonds for Paris, Jungfraujoch and Wollongong, respectively. GEOS-Chem CO total columns are indicated by full circles in black colour."

Page 13, line 354 – Text says satellite data for Jungfraujoch is shown in Figure 3, but so are data for Paris and Wollongong.

- We have added the following information "middle panel".

Page 14, line 384 – due to anthropogenic

- Done.

Page 15, line 401 – what is meant by "warming system"?

- We have modified to "heating system".

Page 16, line 410 – at the end of

- Done.

Page 16, line 411 – at the end of

- Done.

Page 16, line 419 – low-altitude sites

- Done.

Page 16, line 420 – variability to the

- Done.

Page 17, line 434 – The increases after March

- Done.

Page 19, line 484 – which confirm
- Done.

All technical corrections have been taken into account.

We hope that we have answered clearly and in a satisfying manner each of the comments from both referees.

Please also note the supplement to this comment:
http://www.atmos-chem-phys-discuss.net/acp-2015-884/acp-2015-884-AC1-supplement.pdf

**Supplement:**

[revised manuscript text omitted]

---

## Author Response (AR2)

Ref.: ACP-2015-884

We would like to thank the co-editor and both anonymous referees for their reviews and very constructive remarks. These were taken into account in the new version of the paper (online revised paper: "CO_Variability3.pdf").

In the following, we answer to the individual comments and have listed other technical corrections to fix some grammatical errors.

Best regards,
Y. Té

Co-editor

Comment#01

A few small changes could help non experts in FTIR retrievals understand the comparability of these data with surface in situ observations.

Answer#01

We have added the following sentence to justify the comparability between FTIR and in situ observations at page 14, lines 384-385:
    "The comparability between FTIR retrievals and surface in situ observations is thus assured."

Referee#1

Comment#01

First. The use the phrase 'seasonal variation' throughout when 'seasonal cycle' may be more appropriate. The former suggests variation within a season, while the 'season cycle' indicates the annual changes in weather, daylight …, due to yearly variations of the Earth's orbit..

Answer#01

As proposed, we have replaced 'seasonal cycle' by 'seasonal variation' 5 times:
    - Page 1, line 4
    - Page 1, line 14
    - Page 16, line 417
    - Page 18, line 464
    - Page 20, line 484

Comment#02

Second. I would like to see more discussion of the measurements. Not the already detailed technical information, but how differences in instruments effect the results. Comparability (the closeness of agreement between measurements made at the same time and location) of the surface FTR retrievals should be mentioned. Possible biases among instrument results

should be noted. In particular, is there bias resulting from potentially interfering atmospheric constituents with location or seasonality, such a the presence of clouds or aerosols?

Answer#02

We have modified and added some sentences to explain the comparability of the data at page 7, lines 200-207:
  "The following precautions have been taken in order to improve comparability between satellite and ground based measurements:
    - Only clear sky data from ground-based FTIR and satellites has been used.
    - The CO abundance is retrieved from the same spectral domain (4.7 µm for both the ground-based FTIR and the satellite instruments), allowing to minimize possible biases related to spectroscopic parameters and interfering atmospheric constituents (gaseous species, aerosols).
    - Satellite data were selected within a 30 km × 30 km square centered at the site location: corresponding to a range of ±0.15° for the latitude and ±0.23° for the longitude."

Referee#2

Technical Corrections

Page 2, line 49 – From THE end of 1980 until 1997, CO decreased.

- Done at page 2, line 33.

Page 2, line 50 – Since then, a few

- Done at page 2, line 34.

Page 2, line 51 – unusualLY

- Done at page 2, line 35.

Page 3, line 64 – give the Bruker model, as done for the other two sites immediately below

- Done at page 2, lines 47-48: "Bruker IFS 125HR".

Page 7, line 209 – should be "degrees of freedom for signal (DOFS)" here, and DOFS should replace DOF throughout the paper

- Done at different locations:
    - Page 4, lines 108-109: "degrees of freedom for signal (DOFS)".
    - Page 5, line 142: "DOFS"
    - Page 6, line 181: "DOFS"
    - Page 7, line 196: "DOFS"

Page 8, line 240 – which version of WACCM?

- Done at page 5, line 135: "version 4 WACCM".

Page 9, line 209 – fix this sentence

- Done at page 6, line 178: "IASI L2" was removed.

Page 10, line 296 – on-board NASA's Terra …

- Done at page 6, line 186.

Page 15, line 449 – delete colour

- Done at page 10, line 325: we have kept "the".

Page 15, line 456 – interpolated the lower pressure levels ONTO thinner

- Done at page 11, line 335.

Page 17, line 507 – Figure/Fig. 5

- Done at page 14, line 382 and page 15, line 393.

Page 17, line 508 – delete points and point

- Done at page 14, lines 383-384: "two independent measurements of tropospheric CO: the first supplies maximal information".

Page 18, line 2 of caption – comeS

- Done at page 14, line 2 of caption.

Page 19, line 526 – downwind

- Done at page 15, line 399.

Page 20, line 566 – low-altitude

- Done at page 16, line 429.

Page 21, line 1 of caption – Monthly averaged CO in situ …
Page 21, line 2 of caption – green squares for the means of the four urban sites
Page 21, line 3 of caption – is applied to the 5-year monthly means for urban (green line) and mountain (red line) sites.
Page 21, line 4 of caption – The two lowermost datasets show …

- Done at page 17, the caption is now: "Monthly averaged CO in situ measurements at the surface in Switzerland using Swiss NABEL data from 2009 to 2013 (green squares are the means of the four urban sites and red diamonds are for Jungfraujoch). The sine function fit is applied to the 5 years monthly means of urban (green line) and mountain (red line) sites. The two lowermost datasets show the residuals of the fit (green open squares for urban sites and

red small diamonds for Jungfraujoch). Monthly averaged CO surface VMR from GEOS-Chem located at Jungfraujoch is shown in red open circles".

Page 21, line 583 – change close-by to nearby

- Done at page 16, line 440.

Page 24, line 650 – timescale THAN complete

- Done at page 20, line 497.

Page 24, line 640 – state explicitly whether this time lag is seen in the measurements, or in the measurements and GEOS-Chem

- Done at page 20, lines 488-490: "Interestingly, a time-lag of about 2 months between upper altitude and surface CO has been found in the measurements and GEOS-Chem in both Paris and Jungfraujoch".

Page 25, line 655 – instruments are capable of SAMPLING the …

- Done at page 20, line 502.

Page 33, Table 5 caption – Ground-based FTIR instrument parameters for the three measurement stations.

- Done at page 29, the caption is now: "Ground-based FTIR instrument parameters for the three measurement stations".

Below, there are other technical corrections to fix some grammatical errors (mentioned by referee #02): 'The manuscript still has many distracting grammatical errors that should be corrected'

- Page 1, line 1: The → This

- Page 1, line 11: in Paris and at Jungfraujoch → at Paris and Jungfraujoch

- Page 1, line 14: allow identification of → identify

- Page 1, line 15: In Paris and at Jungfraujoch → At both Paris and Jungfraujoch

- Page 1, line 17: In → At

- Page 2, line 24: terpene), which are emitted by plants → terpene, which are emitted by plants)

- Page 2, line 31: Before 1980, there were only few measurements, which → The limited measurements before 1980

- Page 2, line 34: CO has decreased → atmospheric CO decreased

- Page 2, line 39: CO were initiated through → CO began with

- Page 3, lines 66-67: at the East coast of Australia and about 80 km from the South of Sydney. Here, we present NDACC analysis data on → on the East coast of Australia, about 80 km from the south of Sydney. Here, we analyse NDACC data for

- Page 3, lines 72-73: data which → information which

- Page 3, lines 80-81: from the remote sensing data and the one from the surface → from both the remote sensing data and the surface

- Page 4, line 86: on → for

- Page 4, line 88: Using appropriate → Appropriate

- Page 4, line 90: But for the present CO study → For this CO study

- Page 4, lines 106-107: wings of that line → wings of the P(8) line

- Page 5, line 118: thanks to the → due to the

- Page 5, line 128: fitting → fit

- Page 5, line 134: for the → for

- Page 8, lines 226-227: For the paper, we have focussed on the urban sites Bern, Lausanne, Lugano and Zürich as → In this paper, we have focussed on the urban sites Bern, Lausanne, Lugano and Zürich, as

- Page 8, line 230: Results of surface CO at Wollongong were obtained from → Surface CO at Wollongong is measured using

- Page 8, line 244: In situ data was monthly → For this paper, in situ data is monthly

- Page 9, line 258: CO is emitted from anthropogenic, biomass burning and biofuel burning sources → CO is sourced from anthropogenic, biomass burning and biofuel burning emissions

- Page 10, lines 299-301: The CO mean value of $1.1 \times 10^{18}$, molecules/cm$^2$ at Jungfraujoch is quite low due to the → The CO column mean value of $1.1 \times 10^{18}$, molecules/cm$^2$ at Jungfraujoch is quite low and is attributed to the

- Page 10, lines 302-303: For its characterisation, we have used a sine function → We have used a sine function to characterize seasonality

- Page 10, line 307: Here y represents → Where y represents

- Page 10, line 314-315: but not significantly higher, → but not significantly, higher than at Jungfraujoch

- Page 12, line 347: over the → for the

- Page 12, line 5 of the Figure 2 caption: by full circles in black colour → black full circles

- Page 13, line 3 of the Figure 3 caption: blue one are → blue dotted line are

- Pages 12-13, lines 353-355: as compared to the ground-based FTIR stations in the Southern Hemisphere ranging from -19.2% to -27.5% and depending on → when compared to the ground-based FTIR stations in the Southern Hemisphere ranging from -19.2% to -27.5% and concluded differences depend on

- Page 13, line 356: events unaccounted → events are unaccounted

- Page 13, line 361: and not at reproducing → rather than reproducing

- Page 13, line 363: Northern Hemispheric → Northern Hemisphere

- Page 13, line 369: hilly → mountainous

- Page 15, line 412: a maximum → a surface maximum

- Page 16, lines 424-425: Quite differently, in situ surface CO at Jungfraujoch shows the same seasonal variability as the whole atmosphere (characterized by the total column seasonality) being shifted → In contrast, in situ surface CO at Jungfraujoch shows the same seasonal variability as the whole atmosphere (characterized by the total column seasonality) and is shifted

- Page 16, lines 426-428: modeling at Jungfraujoch. Unlike the modeling for Paris where the underestimation is much stronger, the GEOS-Chem underestimates the CO surface VMR by about 23% → model at Jungfraujoch. Unlike the model at Paris, where the underestimation is much stronger, GEOS-Chem underestimates the CO surface VMR by about 23% at Jungfraujoch

- Page 16, lines 434-435: Hemispheric spring (Edwards et al., 2006). Unlike the two Northern Hemispheric sites, there seems to be no → Hemisphere spring (Edwards et al., 2006). Unlike the two Northern Hemisphere sites, there is no

- Page 17, lines 448-449: another three GEOS-Chem simulations have been run → we perform three GEOS-Chem sensitivity simulations

- Page 17, line 450: these runs → these simulations

- Pages 18-19, lines 468-469: Inversely, shutting off either biomass burning or biogenic emissions, only weakly affect the seasonal variation and the maximum peaks. → In comparison, shutting off either biomass burning or biogenic emissions, only weakly affects the seasonal variation and the maximum peaks.

- Page 19, lines 469-470: As compared to the standard run, CO columns are just a little bit lower due to some emissions missing. → Compared to the standard run, CO columns are marginally lower due to some missing emissions.

- Page 19, line 475: Northern. → Northern Hemisphere.

- Page 20, line 490: lag is likely linked to → lag is likely due to

- Page 20, lines 498-500: study more closely the link between local and non-local emission sources and the magnitude of the time shift between surface and total column CO by extending the present study on more sites and to improve the analysis → study the link between local and non-local emission sources and the magnitude of the time shift between surface and total column CO by extending the present study to more sites and improving the analysis

We hope that we have answered clearly and in a satisfying manner each of the comments from the co-editor and both referees.